# Integrated multi-omics analysis of adverse cardiac remodeling and metabolic inflexibility upon ErbB2 and ERRα deficiency

Catherine R. Dufour [1], Hui Xia [1,2], Wafa B'chir[1], Marie-Claude Perry[1,2], Uros Kuzmanov[3], Anastasiia Gainullina[4,5], Kurt Dejgaard[2], Charlotte Scholtes [1], Carlo Ouellet[1], Dongmei Zuo[1], Virginie Sanguin-Gendreau[1], Christina Guluzian[1,2], Harvey W. Smith[1], William J. Muller [1], Etienne Audet-Walsh[1], Alexey A. Sergushichev [4], Andrew Emili[3] & Vincent Giguère [1,2 ✉]

Functional oncogenic links between ErbB2 and ERRα in HER2+ breast cancer patients support a therapeutic benefit of co-targeted therapies. However, ErbB2 and ERRα also play key roles in heart physiology, and this approach could pose a potential liability to cardiovascular health. Herein, using integrated phosphoproteomic, transcriptomic and metabolic profiling, we uncovered molecular mechanisms associated with the adverse remodeling of cardiac functions in mice with combined attenuation of ErbB2 and ERRα activity. Genetic disruption of both effectors results in profound effects on cardiomyocyte architecture, inflammatory response and metabolism, the latter leading to a decrease in fatty acyl-carnitine species further increasing the reliance on glucose as a metabolic fuel, a hallmark of failing hearts. Furthermore, integrated omics signatures of ERRα loss-of-function and doxorubicin treatment exhibit common features of chemotherapeutic cardiotoxicity. These findings thus reveal potential cardiovascular risks in discrete combination therapies in the treatment of breast and other cancers.

[1] Goodman Cancer Institute, McGill University, Montréal, QC H3A 1A3, Canada. [2] Department of Biochemistry, Faculty of Medicine, McGill University, Montréal, QC H3G 1Y6, Canada. [3] Donnelly Centre for Cellular and Biomolecular Research, University of Toronto, Toronto, ON M5S 3E1, Canada. [4] ITMO University, 49 Kronverksky Prospekt, Saint Petersburg 197101, Russia. [5] Koltzov Institute of Developmental Biology, Russian Academy of Sciences, 26 Vavilov Street, Moscow 119334, Russia. ✉email: vincent.giguere@mcgill.ca

The heart relentlessly relies on the coordinated regulation of mitochondria, energy-producing pathways, and signaling conduits to fuel contraction and sustain blood flow, and these functions are dependent on the precise control of metabolite levels, gene expression, and enzyme activities. Any derangement can negatively impact cellular homeostasis and drive pathological remodeling. Heart failure remains an important cause of mortality worldwide posing a tremendous burden on the healthcare system[1]. Characterization of cardiac metabolic, molecular, and structural reprogramming events in response to genetic or extrinsic factors is key to understanding the metabolic flexibilities of the heart, identify causal determinants and risk factors, and ultimately guide drug development and treatment strategies for a wide range of diseases.

The tyrosine kinase receptor ErbB2 (also referred to as HER2) is well-known for its oncogenic activity in breast cancer (BCa), but it also plays an important role in both cardiac embryonic development and in the adult heart[2,3]. Genetic or pharmacological inhibition of ErbB2 has been shown to cause dilated cardiomyopathy (DCM), characterized by chamber dilation and decreased contractility[4–6]. Treatment regimens for HER2+ BCa typically involve ErbB2-targeted therapies including trastuzumab in combination with chemotherapies such as the anthracycline doxorubicin and alkylating agent cyclophosphamide to enhance the anti-tumor effects of HER2-blockade, albeit increasing the cardiotoxic risk from 3-7% to 27% of patients[7–9]. Impaired stress responses and cardiomyocyte apoptosis consequential to compromised cell survival and repair are implicated in trastuzumab-induced cardiac dysfunction[10]. Doxorubicin-induced adverse cardiac effects are the most severe; however, both doxorubicin and cyclophosphamide can cause mitochondrial damage and dysfunction, oxidative and nitrative stress, calcium deregulation, inflammation, and fibrosis[10,11]. The precise underlying mechanisms for the observed cardiotoxicities remain incompletely understood, and inevitably, treatment cessation due to adverse cardiac events is undesirable, warranting further investigation into the identification of underlying risk factors and causal mechanisms.

Orphan nuclear receptor oestrogen-related receptor α (ERRα, NR3B1) is a key transcriptional regulator of mitochondrial function, redox homeostasis and energy metabolism[12–14], thus being an attractive therapeutic target for the treatment of metabolic disorders and diseases including type 2 diabetes, obesity and cancer[15–17]. ERRα is also a broad regulator of cardiac programs including intracellular fuel sensing, fatty acid β-oxidation (FAO), citric acid cycle (CAC), ATP transport, and calcium handling[18]. ERRα is essential for the bioenergetic and functional adaptation to cardiac pressure overload induced by transverse aortic constriction via its direct transcriptional regulation of ATP generating programs[19].

In malignant BCa, ERRα, and ErbB2 are functionally linked and their expression levels correlate positively[20–22]. Notably, attenuation of ERBB2 signaling disrupts ERRα activity[22], and reciprocally, ERRα ablation reduces ERBB2 amplicon gene transcription and impedes ErbB2-induced murine BCa development[20]. Thus, targeting ERRα in combination with ErbB2 may offer a therapeutic benefit in HER2+ patients[23]. However, as both factors play cardioprotective roles, we investigated the consequence of their combined loss of function on the heart. Herein, we report that in-depth cross-analyses of cardiac phosphoproteomic, transcriptomic and metabolic profiles reveal that while ErbB2 and ERRα play distinct and complementary roles in maintaining myocardial homeostasis and function, combined genetic attenuation of ErbB2 and ERRα severely amplifies adverse cardiac remodeling and metabolic inflexibility observed in mice deficient in a single factor. Furthermore, an integrated omics

signature driven by ERRα loss was predictive of doxorubicin response and reciprocally, an assembled cardiac multi-omics doxorubicin signature was found to be characteristic of decreased ERRα activity, identifying a hitherto unsuspected functional link between ERRα and doxorubicin action in the heart.

## Results

**Loss of ErbB2 and ERRα signaling independently contribute to myocardial dysfunction.** To investigate the functional consequence and possible molecular and genetic crosstalk between ErbB2 and ERRα in the adult heart, we crossed ERRα knock-out (KO) mice[24] with the ErbB2 hypomorphic mouse model[25] (ErbB2 KI) giving rise to ErbB2 KI/ERRα KO mice (herein referred to as KI:KO) in a FVB background. KI:KO mice are viable, fertile and do not display any gross anatomical abnormalities. The concomitant loss of both ErbB2 and ERRα in KI:KO mice was confirmed by RT-qPCR (Supplementary Fig. 1a). Given that ErbB2 KI mice develop age-dependent DCM with earliest signs of pathophysiology at 4-months of age[6], cardiac function was evaluated on 15-week-old ErbB2 KI, ERRα KO, and KI:KO mice in comparison to WT controls.

Loss of ErbB2 and ERRα had opposite effects on heart size with KI:KO reflecting the average outcome (Fig. 1a–c). Masson's trichrome staining revealed increased fibrosis in ERRα KO hearts and to a greater extent in KI:KO mice (Fig. 1a, d). Ultrasound echocardiography confirmed the development of DCM in the ErbB2 KI model, marked by a reduction in left ventricular (LV) contractile function as demonstrated by lower LV ejection fraction (LVEF) and LV fractional shortening (LVFS) parameters as well as increased LV dilatation with augmented end-systolic (LVIDs) and end-diastolic dimensions (LVIDd) relative to WT (Fig. 1e–i). ERRα KO mice also presented with DCM similarly to ErbB2 KI mice (Fig. 1e–i), with cardiac fibrogenesis likely playing a crucial role in the pathogenesis. KI:KO displayed a synergistic effect of impaired ErbB2 and ERRα signaling on DCM development (Fig. 1e–i), not associated with greater vascular defects, cardiomyocyte apoptosis or hypertrophy (Supplementary Fig. 1b–e). Consistent with their increased disease severity, KI:KO hearts expressed higher transcript levels of two biomarkers of hemodynamic stress and heart failure, atrial natriuretic peptide (ANP) and brain natriuretic peptide (BNP)[26] (Fig. 1j), thus supporting both ERRα- and ErbB2-dependent contributions to the observed DCM.

**Cardiac phosphoproteomics reveals ErbB2 and ERRα dependencies for proper structural and functional integrity.** Changes in protein phosphorylation and associated signaling mechanisms have been linked with cardiac dysfunction. Unbiased label-free LC-MS/MS-based phosphoproteomics profiling identified 2066 phosphorylated peptides mapped to 602 proteins using a high confidence site localization probability of the modified residues ($\geq 0.7$). Relative to WT, KI:KO hearts displayed 48 to 63% more significant phosphopeptide changes (limma, $p < 0.05$, $|FC| \geq 1.5$) than ErbB2 KI and ERRα KO models, respectively, mostly serine modifications, and clustered more closely with ErbB2 KI samples (Fig. 2a, Supplementary Fig. 2a–c, and Supplementary Data 1). Consolidated phosphoserine and phosphothreonine motifs of differentially expressed phosphopeptides were marked by a strong preference for proline at position +1 and arginine at position −3 in all groups compared to WT (Fig. 2b and Supplementary Fig. 2d). The MoMo tool[27], which expands on the robust algorithm Motif-x[28], identified the pSP and RXXpS motifs in all 3 models vs WT, the former being the most frequently observed motif (Fig. 2c and Supplementary Fig. 2e). Among the other enriched motifs identified, we found the pTP motif in both ERRα

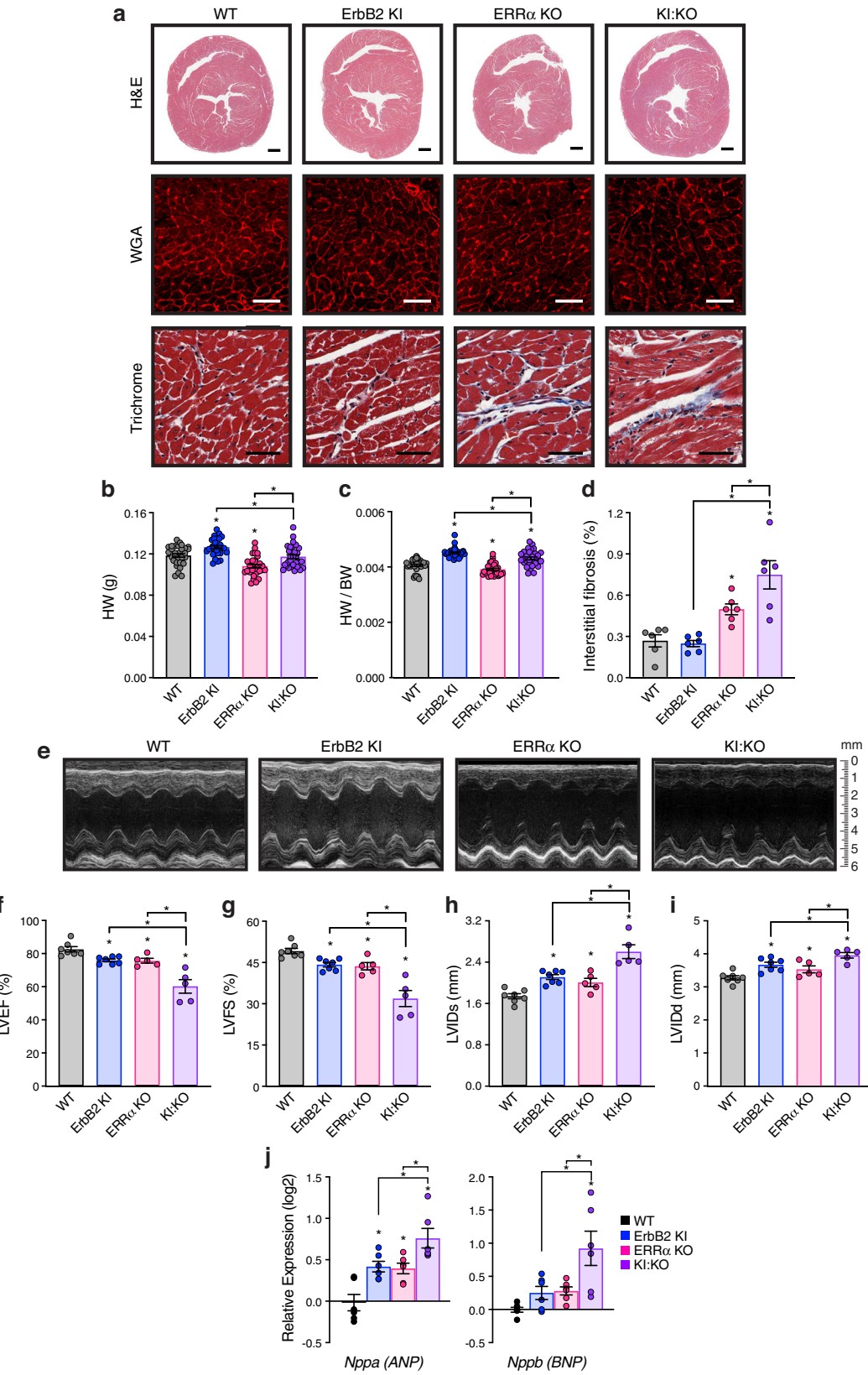

KO and KI:KO as well as pSXXE and RXXpT motifs in the KI:KO model. PhosphoMotif Finder[29] predicted GSK3 and ERK family kinases to be responsible for a significant number of altered phosphopeptides along with CaMKII, PKA, and PKC (Fig. 2c).

Kinase Enrichment Analysis 2 (KEA2)[30] found altered phosphorylation states of gap junction protein alpha 1 (Gja1),

also known as connexin 43 (Cx43), among several biological terms (e.g., gap, ion-channel gating, trafficking, junctions, conductance) linked to diminished ErbB2 signaling (Supplementary Fig. 2f). Metabolism-related terms PDK1 (pyruvate dehydrogenase kinase 1) and insulin associated with ERRα loss. GO cellular component enrichment analysis of differentially

**Fig. 1 Cardiac phenotype of mice with impaired ErbB2 and/or ERRα activity. a** Representative Hematoxylin and Eosin (H&E), wheat germ agglutinin (WGA), and Masson's trichrome staining of heart sections from 15-week old mice with ErbB2 and/or ERRα loss-of-function. Scale bars, 500 μm (H&E) and 50 μm (WGA, Trichrome). **b, c** Mean heart weight (HW) (**b**) and normalized HW to body weight (BW) ratios (**c**) of mice ($n = 30$). **d** Quantification of interstitial fibrosis (Trichrome, $n = 6$). **e** Representative M-mode echocardiographic images for each mouse genotype. **f–i** Percent of left ventricular ejection fraction (LVEF) (**f**) and LV fractional shortening (LVFS) (**g**) as well as echo-derived LV internal diameter end systole (LVIDs) (**h**) and end diastole (LVIDd) (**i**) in mice (WT and ErbB2 KI, $n = 7$; ERRα KO and KI:KO, $n = 5$). Cardiac RT-qPCR analysis of the genes encoding atrial natriuretic peptide (ANP) and brain natriuretic peptide (BNP), both markers of cardiac dysfunction (**j**) ($n = 6$). Data are normalized to Rplp0 levels. Data in (**b–d**) and (**f–j**) represent means ± SEM, *$p < 0.05$ by ANOVA relative to WT controls, unless otherwise indicated. See also Supplementary Fig. 1.

expressed phosphoproteins (DEPPs) found actin cytoskeleton, focal adhesion, intercalated disc, and the contractile apparatus among the top 10 targets across the models (Fig. 2d, Supplementary Fig. 2g, and Supplementary Data 1). Ingenuity Pathway Analysis (IPA) revealed PKA, DCM and integrin linked kinase (ILK) among common over-represented pathways in the 3 groups compared to WT, with a greater number of associated perturbations found in KI:KO hearts (Supplementary Fig. 2h and Supplementary Data 1).

The cardiomyocyte localization of a subset (51) of KI:KO DEPPs is illustrated in Fig. 2e with selected ErbB2 and/or ERRα-dependent effects shown in Fig. 2f. The data indicate deregulation of protein activities at all three cell-cell junctions in intercalated disks including Cx43 (S306, S314, S328, S364, S365, S368) at gap junctions, desmoplakin (Dsp; S2, S7, S2221) at desmosomes, and catenin alpha 3 (Ctnna3; S637, S647, T649) part of fascia adherens. Immunoblot analysis of phospho-Cx43 S368 levels support the hypophosphorylation of gap junction proteins in hearts with down-regulated ErbB2 signaling (Fig. 2g). Sarcoplasmic reticulum proteins, important for calcium signaling and contraction, as well as sarcomeric proteins exhibited noted phospho-levels changes. In addition, DEPPs involved in anchoring sarcomeres to the costameric complexes and plasma membrane via the actin cytoskeleton were found including vinculin (Vcl; S97, S346), desmin (Des; S28, S32, S68), filamin C (Flnc; S2234, S2237) and dystrophin (Dmd; S3616).

A limited number of DEPPs involved in cellular metabolism were found, many of which were linked to glucose metabolism (Fig. 2h). In particular, hyperphosphorylation of the rate-limiting glycolytic enzyme 6-phosphofructo-2-kinase/fructose-2,6-biphosphatase 2 (Pfkfb2) at the known activating site S486 was observed in all models compared to WT, indicating their increased reliance on glucose. Phosphorylation of Aldoa at S36, found significantly elevated in ERRα KO and more prominently in KI:KO hearts, was recently found to drive glycolytic metabolism of liver cancer cells[31]. In the absence of ERRα, phosphorylation at the inhibitory site S232 of the E1 alpha 1 subunit (Pdha1) of the pyruvate dehydrogenase (PDH) complex was strongly reduced, an effect confirmed by immunoblot analysis (Fig. 2h, i). As indicated by KEA2 (Supplementary Fig. 2f), this finding is likely attributable to decreased expression of the upstream kinase Pdk1 (Fig. 2i). Consistently, increased PDH activity was found in ERRα KO and KI:KO hearts (Fig. 2j). Taken together, ERRα and ErbB2 contribute to the phospho-level regulation of proteins associated with structural integrity, contractile force, and glucose handling.

**Transcriptomics identifies ERRα-dependent signatures of inflammation and fibrogenesis.** Cardiac transcriptome profiling using DNA microarray technology revealed a considerably greater number of differentially expressed genes (DEGs) in KI:KO hearts relative to WT, overlapping 38% and 8% with ERRα KO and ErbB2 KI groups, respectively (ANOVA, $p < 0.05$, $|FC| \geq 1.2$) (Fig. 3a, Supplementary Fig. 3a, and Supplementary Data 2). As such, a larger subset of KI:KO DEGs were linked to cardiac

disease phenotypes (Supplementary Fig. 3b and Supplementary Data 2). Similar to phosphoproteomics analyses (Supplementary Fig. 2g), GO enrichment component analysis showed a stronger enrichment of DEGs related to cardiomyocyte structural framework in KI:KO hearts, specifically the intercalated disk, actin cytoskeleton and focal adhesion (Supplementary Fig. 3c). Cross-examination of DEGs and DEPPs showed marginal overlap in ErbB2 KI with ~20% of ERRα KO and KI:KO DEPPs displaying altered transcriptional regulation (Supplementary Fig. 3d). The data support operative transcriptional and post-translational mechanisms underlying the pathological cardiac remodeling.

Gene Set Enrichment Analysis uncovered a significant up-regulation of inflammation-related gene signatures in ERRα KO and KI:KO hearts (Fig. 3b and Supplementary Data 2). In agreement, ERRα loss triggered macrophage infiltration (Fig. 3c), noting a general switch from pro-inflammatory M1-type to an anti-inflammatory and pro-fibrotic M2-type macrophage gene expression profile (Supplementary Fig. 3e). ERRα KO and KI:KO hearts displayed the highest expression of matrisome genes encoding extracellular matrix (ECM) and ECM-associated proteins including key genes involved in transforming growth factor β (TGF-β) activation and signaling (Fig. 3d, e). TGF-β signaling is known to promote ECM deposition and fibrosis associated with tissue inflammation and injury[32]. Indeed, Masson's trichrome showed augmented fibrogenesis in ERRα KO and KI:KO hearts (Fig. 1a, d), further supported by their elevated expression of α-smooth muscle actin (α-SMA) encoded by Acta2 (Fig. 3f, g and Supplementary Data 2).

**Computational identification of cardiac metabolic gene modules.** To establish a regulatory link between ErbB2 and ERRα and cardiac metabolism, we applied an in-house bioinformatics tool capable of identifying networks of interconnected transcriptionally-regulated metabolic genes (modules) capable of multi-condition comparisons[33–35]. Our analysis discovered seven differentially regulated modules (Fig. 4a). Impaired ErbB2-signaling alone had minimal impact but essentially neutralized the effects of ERRα inhibition on modules II and V (Fig. 4a and Supplementary Fig. 4). In stark contrast, concomitant loss of ErbB2 and ERRα significantly perturbed five modules (I, III, IV, VI, VII), with modules I and VI largely ascribed to ERRα loss alone (Fig. 4a–c). Interconnected gene networks involved in FA degradation, amino acid, pyruvate, CAC, and phospholipid metabolism were most significantly down-regulated in KI:KO hearts (Fig. 4b). On the other hand, a more prominent up-regulation of networks linked to N-glycan biosynthesis, glycolysis, arachidonic acid, phospholipid, and sphingolipid metabolism was found in these mice (Fig. 4c). Phospholipid metabolism was paradoxically found as both activated and repressed.

**Cardiac metabolomics reveals ERRα- and ErbB2-dependent metabolic reprogramming.** To validate the power of our computational tool to predict the dysregulation of precise metabolic programs, we proceeded to characterize the mouse cardiac

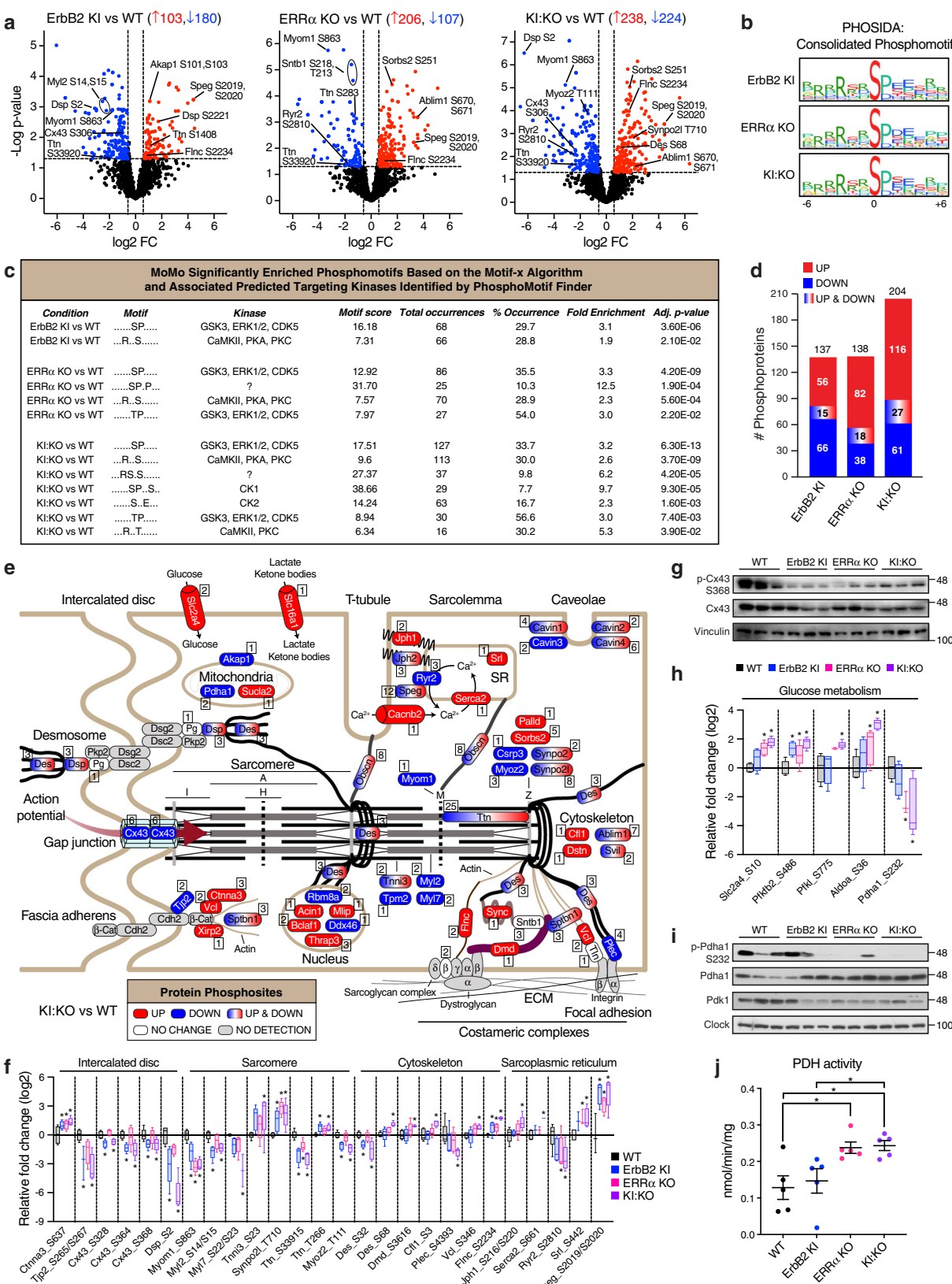

metabolomes. Cardiac large-scale UPLC/MS-based untargeted profiling resulted in the detection of 571 biochemicals (Supplementary Data 3). KI:KO hearts displayed 287 differentially expressed metabolites (DEMs; ANOVA, p < 0.05) relative to WT, that were primarily down-regulated, mostly ERRα-dependent, and largely related to lipid metabolism (Fig. 5a, b). Supervised

clustering using partial least squares-discriminant analysis (PLS-DA) showed a similar group separation as to unsupervised clustering (Fig. 5b), highlighting ADP-ribose, pyruvate and AICAR as the most distinguishing metabolites (strongest VIP scores), though 40% of the top 15 were annotated to lipid metabolism (e.g., acetyl-CoA, corticosterone) (Supplementary

**Fig. 2 Phosphoproteomics identification of ErbB2 and ERRα post-translational control of cardiomyocyte structure and metabolism. a** Volcano plots illustrating the significantly up-regulated (red) and down-regulated (blue) phosphopeptides from cardiac phosphoproteomics profiling of mouse models relative to WT (limma, $p < 0.05$, $|FC| \geq 1.5$, $n = 5$). **b** Consolidated phosphomotifs generated by PHOSIDA[79] of pSer-modified phosphopeptides found differentially expressed in the mouse models compared to WT showing site-specific amino acid preferences adjacent to the central serine phosphorylated residue. **c** Phosphopeptide sequence motif discovery by the MoMo[27] software tool based on the Motif-x[28] algorithm along with PhosphoMotif Finder[29] predicted phosphomotif-targeting kinases. **d** Bar chart showing the number of identified DEPPs from (**a**) harboring up- and/or down-regulated site-specific phosphorylation (limma, $p < 0.05$, $|FC| \geq 1.5$, $n = 5$). **e** Schematic showing the cardiomyocyte localization of 25% of DEPPs identified in KI:KO hearts from (**d**). Both the number and directional change of protein phosphosites are indicated. Sarcomere regions are shown: M, M line; Z, Z line; I, I band; A, A band, H, H zone. **f** Box plot of relative phosphorylation levels of DEPPs important for cardiomyocyte structural integrity across the genotypes relative to WT ($n = 5$). **g** Immunoblot analysis of total and phospho-Cx43 (S368) levels in heart tissue extracts ($n = 3$). Vinculin levels are shown as a loading control. **h** Box plot of relative phosphorylation levels of DEPPs important for glucose uptake and catabolism across the genotypes relative to WT ($n = 5$). **i** Immunoblot analysis of total and phospho-Pdha1 (S232) levels and its upstream kinase Pdk1 in heart tissue extracts ($n = 3$). Clock levels are shown as a loading control. **j** Scatter dot plot of mouse cardiac pyruvate dehydrogenase (PDH) activities ($n = 5$). Data in (**f**) and (**h**) are shown as box and whiskers plots: center line denotes median, box extends from 25th to 75th percentiles, and whiskers extend to the lowest and highest values; *$p < 0.05$, $|FC| \geq 1.5$ by limma relative to WT controls. Data in (**j**) represent means ± SEM; *$p < 0.05$ by ANOVA. See also Supplementary Fig. 2.

Fig. 5a). Random Forest classification differentiated all four groups with an overall predictive accuracy of ~88%, substantially greater than random chance alone (25% accuracy), with most top classifying metabolites related to lipid metabolism including myo-inositol, L-carnitine, and phospholipid species (Supplementary Fig. 5b).

Impaired ErbB2 signaling modestly impacted FA levels, mostly poly-unsaturated FAs (PUFAs), whereas ERRα ablation reduced medium- and long-chain FAs along with PUFAs including linoleic and arachidonic acid (Fig. 5c). Diminished circulating FA pools mirrored cardiac FA profiles (Supplementary Fig. 6a). ERRα loss significantly lowered arachidonic acid-derived eicosanoids with lowest levels in KI:KO hearts (Supplementary Fig. 6b). These findings contrasted with the up-regulation of two metabolic modules encompassing arachidonic acid metabolism in KI:KO, driven by increased transcription of cyclooxygenase-1 (COX-1) and COX-2-encoding genes, *Ptgs1* and *Ptgs2*, respectively, required for prostaglandin synthesis from arachidonic acid (Fig. 4c). Higher COX-2 protein levels in KI:KO hearts were validated by immunoblot analysis (Supplementary Fig. 6c). Although the prostaglandin subclass of eicosanoids was not measured in our metabolomics study, the data indicate their augmentation in KI:KO mice. Further, ERRα inhibition promoted the biosynthesis of phosphatidylcholine (PC) and phosphatidylethanolamine (PE) phospholipids, while a general marked decline in PC- and PE-containing glycerophospholipids, lysolipids, plasmalogens and lysoplasmalogens was observed (Supplementary Fig. 6d–f and Supplementary Data 3). These data corroborate the inferred bi-directional transcriptional control of phospholipid metabolism (Fig. 4b, c). ERRα deficiency also stimulated sphingomyelin production (Supplementary Fig. 6g), supported by up-regulation of sphingolipid metabolic genes (Fig. 4c). Remarkably, while several fatty-acyl carnitine species were decreased in ERRα KO hearts, dual inhibition of ERRα and ErbB2 led to their robust global decline (Fig. 5d). KI:KO also displayed the highest and lowest levels of free CoA and carnitine, respectively (Fig. 5e and Supplementary Fig. 6h).

Disturbances in carbohydrate metabolism were found, with more prominent elevations in glycogen metabolites and many glycolytic intermediates in KI:KO (Fig. 5f). Albeit increased, several metabolites including glucose did not reach statistical significance, likely due to the non-static flux of glucose through the glycolytic pathway. Higher myocardial and serum levels of lactate were observed in conjunction with increased cardiac mRNA expression of lactate dehydrogenase alpha (Ldha) (Fig. 5f–h). Hearts lacking ERRα exhibited markedly diminished pyruvate levels with accumulating upstream glycolytic intermediates found more striking in KI:KO (Fig. 5f). While the

activity of pyruvate kinase was not perturbed (Supplementary Fig. 6i), ERRα KO and KI:KO hearts exhibited augmented PDH activities (Fig. 2j). This significant up-regulation of pyruvate oxidation and pattern of upstream glycolytic intermediates denotes a "bottleneck" of pyruvate entry into the CAC in the absence of ERRα.

Regarding amino acid metabolism, loss of ErbB2 alone resulted in either no change or an increase in a few amino acids (e.g., alanine and glutamine) in stark contrast to ERRα inhibition which generally lowered amino acid profiles (Fig. 5i). Glutamate accumulation was a common feature, a precursor for glutathione (GSH) synthesis important for antioxidant response, also found augmented (Supplementary Fig. 6j). Dipeptide levels (Fig. 5j), an indicator of protein synthesis, reflected the mouse heart to body weight ratios (Fig. 1c).

FA, glucose, and ketogenic amino acid oxidation each contribute to the intracellular acetyl-CoA pool, found independently fostered by impaired ErbB2 and ERRα signaling, denoting decreased incorporation into the CAC (Fig. 5k). Several perturbations in CAC intermediates were observed including increased succinate and decreased fumarate levels in KI:KO (Supplementary Fig. 6k). In addition, loss of either factor augmented α-ketoglutarate derivative 2-hydroxyglutarate (2-HG), found highest in KI:KO hearts (Supplementary Fig. 6k).

**Integrative omics analysis of ErbB2- and ERRα-dependent signatures.** To gain a more holistic understanding of ErbB2 and ERRα contributions to increased DCM severity in KI:KO mice, integrated omics signatures comprising molecules with a greater likelihood for disease causality were generated (Fig. 6a and Supplementary Data 4). Perturbations occasioned by the loss of one factor and counterbalanced by loss of the other were deemed unlikely to be causal. Multi-omics signatures of ErbB2 KI and ERRα KO effects sustained in KI:KO were constructed alongside a KI:KO only signature requiring the concomitant loss of both factors. Pathway enrichment analysis and prediction of pathway activation (z-score $\geq 2$) or repression (z-score $\leq -2$) states were assessed by IPA (Fig. 6b, Supplementary Fig. 7a, and Supplementary Data 4). The ErbB2 KI-driven signature, harboring the least molecules, showed less modulated pathways underscored by phosphoprotein- and/or gene-level changes including perturbed phosphorylation of sarcomeric Myl2 and Tnni3 proteins associated with ILK, PKA and/or DCM signaling and deregulated expression of several growth factors (e.g., *Fgf6/16*, *Igf1*, *Pgf*). S15 and S14/S15 of myosin light chain 2 (Myl2) in human and mouse, respectively, are critical regulatory phosphorylation sites targeted by myosin light chain kinase in which their reduced phosphorylation associates with DCM and heart failure via impacts on

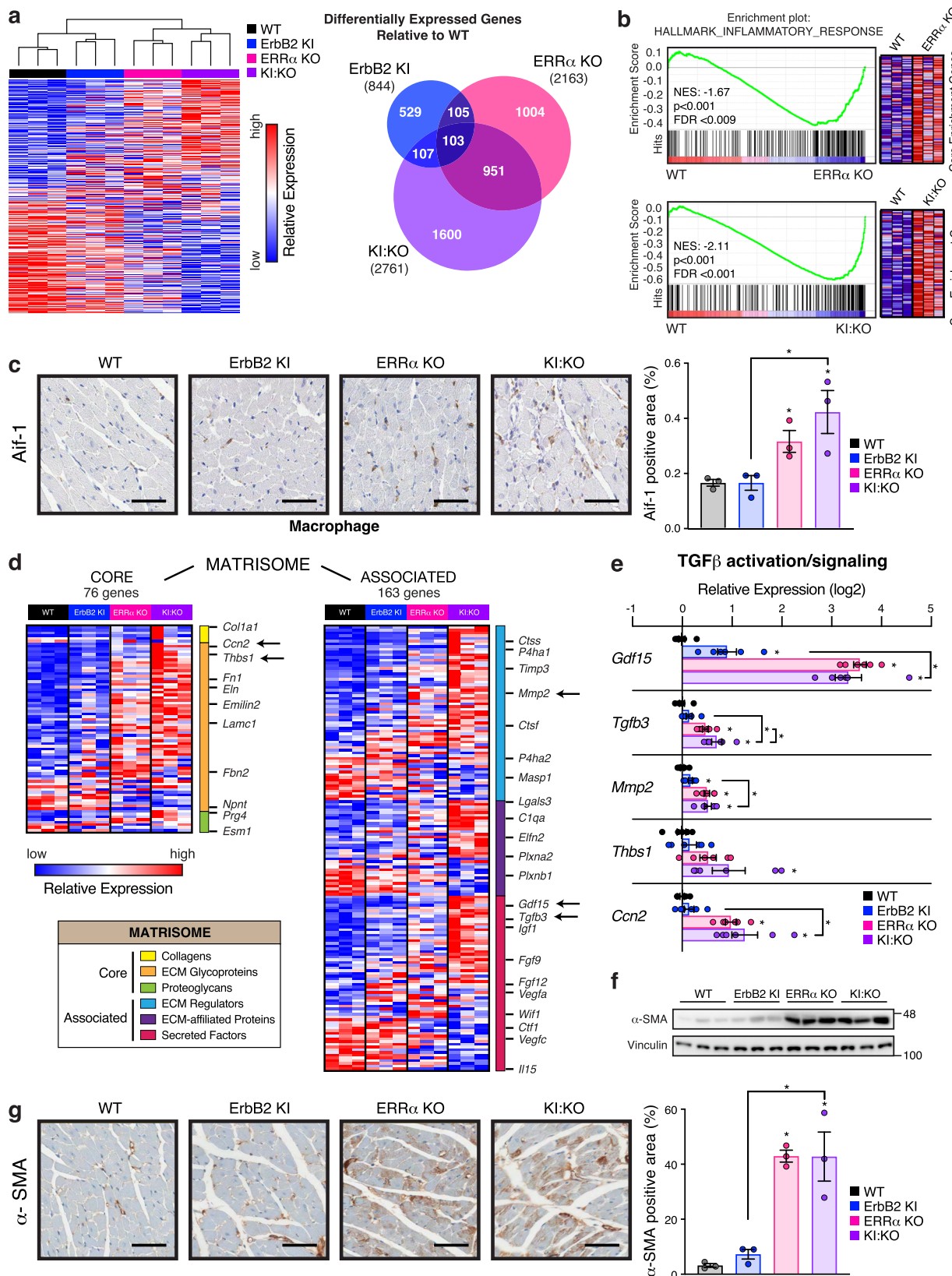

cross-bridge cycling kinetics and contractile force[36]. In support of Myl2 S14/S15 dephosphorylation in hearts with down-regulated ErbB2 activity, we detected lower levels of p-Myl2 using a phospho S15-specific antibody (Supplementary Fig. 7b). Accumulation of protein aggregates marked by hyperphosphorylation of Tau is a well-known hallmark of Alzheimer's disease, but it has also been linked to the pathogenesis of heart disease[37]. Using an antibody recognizing phospho-Tau S385 in mice, corresponding to S396 in human, we observed lower levels of phosphorylated Tau in hearts lacking ErbB2 and/or ERRα, suggesting that this proteotoxic stress is not an underlying cause of the observed DCM in these mice (Supplementary Fig. 7b). The ERRα KO-

**Fig. 3 ERRα ablation in mice induces a transcriptional inflammatory and fibrogenic response. a** Heatmap and Venn diagram illustrating cardiac DEGs (ANOVA, $p < 0.05$, $|FC| \geq 1.2$, $n = 3$) identified across the genotypes relative to WT by microarray analysis. For the heatmap, DEGs were first sorted from most up-regulated to most down-regulated in KI:KO hearts compared to WT prior to unsupervised sample clustering using Euclidean distance measure and average linkage. **b** Gene Set Enrichment Analysis showing an up-regulated hallmark inflammatory gene signature in ERRα KO and KI:KO cardiac transcriptomes vs WT. Normalized enrichment scores (NES), representative plots and gene heatmaps are shown. **c** Representative images and quantification of Aif-1 staining of mouse heart sections as a marker of macrophage infiltration ($n = 3$). Scale bar, 50 μm. **d** Heatmap of matrisome DEGs. Arrows adjacent to genes signify that their expression was validated by RT-qPCR in (**e**). **e** Cardiac RT-qPCR analysis of key genes in (**d**) linked to TGF-β activation and signaling ($n = 6$). Data are normalized to *Rplp0* levels. **f** Immunoblot analysis of α-SMA in heart tissue extracts ($n = 3$). Vinculin levels are shown as a loading control. **g** Representative images and quantification of α-SMA staining of mouse heart sections ($n = 3$). Scale bar, 50 μm. Data in (**c**, **e**, and **g**) represent means ± SEM; *$p < 0.05$ by ANOVA relative to WT controls, unless otherwise indicated. See also Supplementary Fig. 3.

driven signature was marked by the up-regulation of fibrosis signaling (e.g., *Acta2*, *Ccn2*, *Tgfb3*), validated previously (Figs. 1a, d, and 3e–g), as well as perturbations in metabolic processes, noting the transcriptional down-regulation of several mitochondrial FAO (e.g., *Acaa2*, *Acadm*, *Acsl1*) and Oxphos genes (e.g., *Atp5e*, *Ndufb2*, *Ndufv3*). Diminished protein levels of Acadm and Acsl1 in hearts lacking ERRα supported their transcriptional repression with further analysis revealing reduced cardiac mtDNA abundance in ERRα KO and KI:KO mice, consistent with lower expression levels of the mitochondrial transcription factor A (Tfam) (Supplementary Fig. 7c, d). The largest signature, KI:KO only, displayed an up-regulation of molecules associated with immune/inflammation (e.g., *Mapk11*, *Myd88*, *Stat3*) as well as multiple changes linked to cardiac remodeling including altered ILK, PI3K/AKT, PKA, RHOA, ERK/MAPK, mTOR, and actin cytoskeleton. Examination by Western blotting showed increased phosphorylation and hyperactivation of AKT (S473 and T308) and ERK1/2 (T202 and Y204) signaling as well as a prominent elevation in S3 phospho-Cofilin 1 (Cfl1) linked to ILK, RHOA, and actin cytoskeleton (Supplementary Fig. 7e). Cofilins 1 and 2 (Cfl1/2), which functionally overlap, play a key role in sarcomeric actin dynamics and their deactivation by phosphorylation at the conserved S3 residue inhibits their actin severing function, thus impairing actin turnover. Notably, increased Cfl2 S3 phosphorylation was previously observed in human idiopathic DCM patients[38].

Given that mitochondria are a primary target of doxorubicin action, that loss of ERRα triggers mitochondrial dysfunction, and that both facets augment cardiotoxicity induced by ErbB2-targeted approaches, we surmised a strong inverse relationship between doxorubicin and ERRα signaling. Fittingly, a doxorubicin response was deemed activated exclusively in the ERRα KO-driven signature (Fig. 6c), suggesting that impaired ERRα function is a molecular facet of doxorubicin-driven cardiotoxicity. To explore this notion, we constructed a cardiac-specific doxorubicin-dependent multi-omics signature comprising 724 molecules established from 1 phosphoproteomics[39], 9 transcriptomics[40–45] and 9 metabolomics[46–54] datasets derived from doxorubicin-treated animals (Fig. 6d and Supplementary Data 5). Genes/metabolites with inconsistent directional changes across studies were removed and only DEGs found in 5 of 9 transcriptomes were retained. Noteworthy, *Esrra*, found down-regulated in 4 of 9 datasets, did not meet our inclusion criteria. Functional analysis showed enrichment and/or deregulation of pathways linked to doxorubicin including oxidative stress, antioxidant response, ferroptosis, fibrosis, autophagy, and cardiac and metabolic remodeling[55–58] (Supplementary Fig. 7f, g and Supplementary Data 5). Importantly, ERRα activity was strongly down-regulated in the doxorubicin signature (Fig. 6e). Independent analysis of the integrated doxorubicin-modulated transcriptomes, overlapping ~10–25% with respect to the ERRα KO & KI:KO signature, revealed significantly decreased ERRα activity in 7 of 9 datasets (Fig. 6f and Supplementary Fig. 7h). Remarkably,

ERRα was the most consistently and strongly down-regulated transcriptional regulator across the datasets (Supplementary Fig. 7i). Cross-examination of ERRα- and doxorubicin-dependent signatures identified 89 similarly deregulated molecules, 74 genes and 15 metabolites, linked to multiple functions most notably lipid metabolism (Fig. 6g and Supplementary Data 5). Decreased FAs (palmitoleate, linoleate, arachidonate) and FAO-associated *Acsl1*, *Hadh*, and carnitine as well as perturbed membrane lipids were noted. Key targeted processes included p53 (↑*Fas*), insulin (↑*Irs2*), PKA (↓*Pkia*), and RAS signaling (↓*Rasgrp3*), ECM (e.g,. ↑*Ccn2*, ↑*Mmp2*, ↓*Itgb6*, ↓*Sema3b*), cytoskeleton (e.g., ↑*Rhobtb1*, ↓*Auts2*, ↓*Lmod3*), immune/inflammation (e.g., ↑*C4b*, ↑*Osmr*), axon guidance (e.g., ↓*Nrp1*, ↓*Nrp2*), transcription (↑*Per2*, ↑*Tsc22d3*, ↓*Mycn*, ↓*Rcor2*), ATP homeostasis (↓*Adk*, ↓*Rnf207*), oxidative stress response (e.g., ↑*Nfe2l2*, ↑*Hmox1*), calcium homeostasis (↑*Grina*, ↑*Tmbim1*), and iron metabolism/homeostasis (↓*Glrx5*, ↓*Hjv*). These findings establish a previously unknown molecular connection between doxorubicin and loss of ERRα activity in the heart.

## Discussion

The limited access to human heart tissue specimens increases the reliance on animal model systems to study cardiovascular disorders. Herein, we used genetically engineered whole-body mouse models to mimic systemic exposure to candidate drug therapies against BCa. Using a systems biology approach, we characterized ErbB2- and ERRα-dependent molecular features of the DCM developed in these mice whereby their dual inhibition worsened the disease. Our integrated phosphoproteomics, transcriptomics, and metabolomics studies uncovered profound effects of concomitant ErbB2 and ERRα loss-of-function on cardiomyocyte architecture, inflammatory response, and metabolism. To the best of our knowledge, no prior investigation globally mapped the ERRα-dependent phosphoproteome or metabolome in any cellular context and no prior study performed phosphoproteomics or metabolomics profiling downstream loss of ErbB2 signaling in the adult heart. Moreover, integrated multi-omics signatures of ERRα loss-of-function and doxorubicin treatment revealed common features of chemotherapeutic cardiotoxicity. Taken together, our findings highlight previously unrecognized potential cardiovascular risks associated with current and prospective combination therapeutic approaches for the management of BCa.

This study allowed for the delineation of ERRα- and ErbB2-dependent molecular mechanisms. First, reduced ErbB2 signaling lowered PUFAs, up-regulated glycolysis, and altered growth hormone and DCM signaling. Proteins at cell-cell junctions (e.g., Cx43) and within distinct regions of the sarcomere were major targets for altered phospho-dependent regulation, implying that DCM occasioned by ErbB2 inhibition involves decreased mechanical coupling and electrophysiological properties as well as sarcomeric dysfunction via altered structural assembly and force. Notably, our results uncover ErbB2 signaling as a positive regulator of sarcomeric protein Myl2 S14/S15 phosphorylation,

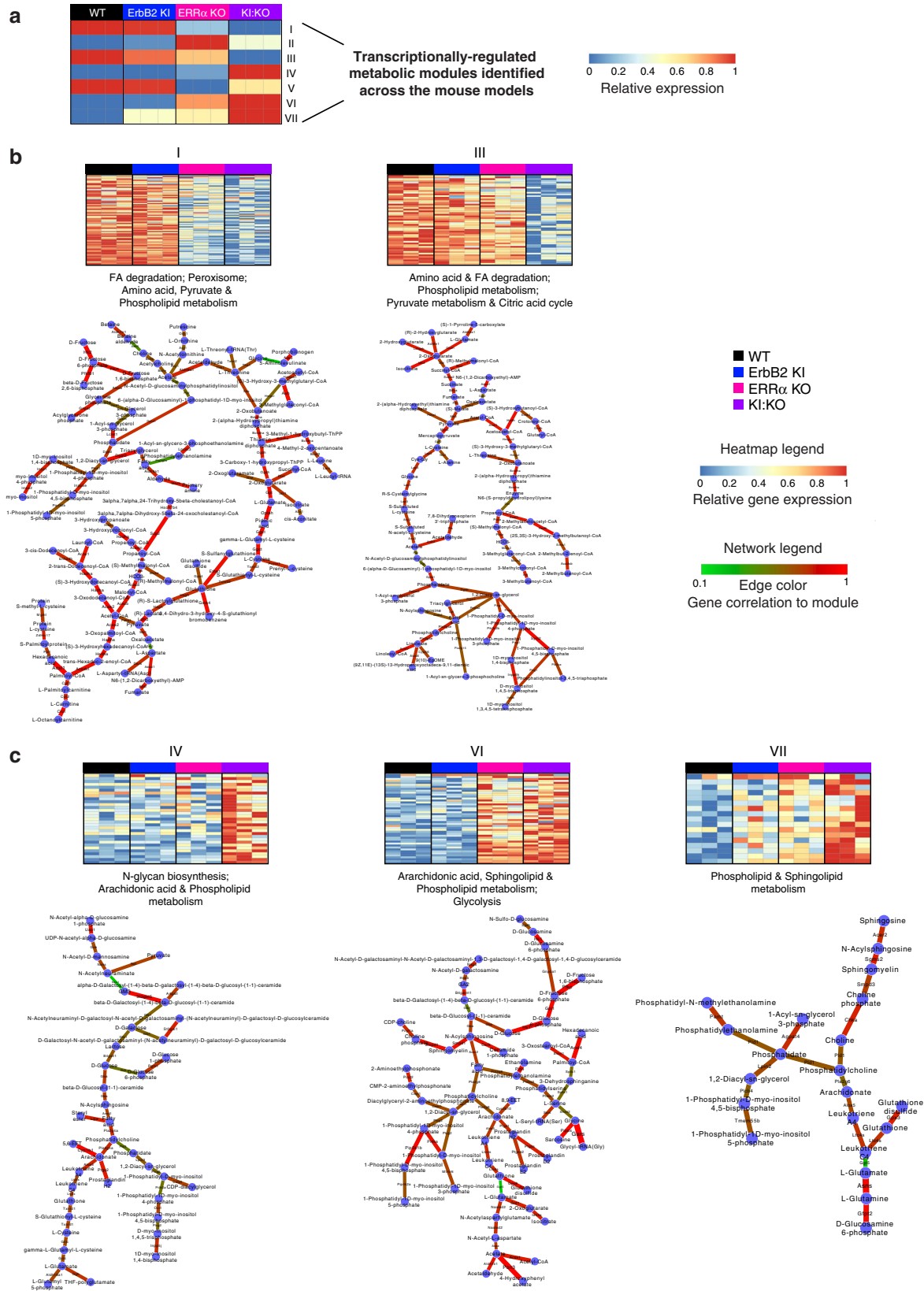

key regulatory sites important for cardiac muscle contraction. The phosphorylation of intercalated disc protein Cx43 at residues S306, S328, S364, S365, and S368 have reported positive actions on gap junction trafficking/assembly and gating[59,60], all found hypophosphorylated in hearts with suppressed ErbB2 signaling. Second, ERRα ablation impacted several signaling processes (e.g.,

DCM, ILK, PKA), fibrosis, immune/inflammation, and reprogrammed energy metabolism. The strong inflammatory signature, ECM deposition, and observed macrophage infiltration further solidify the important role of ERRα in attenuating inflammatory responses found in other tissue/cellular contexts[61–63]. Metabolomics corroborated key transcriptome-based predicted metabolic

**Fig. 4 Computational integrative analysis of mouse heart gene expression profiles and metabolic networks. a** Simultaneous multi-comparison analysis of mouse heart transcriptomes integrated into metabolic networks identified 7 significantly perturbed metabolic modules among the genotypes. Genes found with the greatest correlation in expression and distance in a metabolic reaction network are scored higher and the relative expression of metabolic modules identified for a pattern of scoring genes was calculated across the groups. Heatmap expression patterns for each metabolic module across the groups are shown. **b**, **c** Gene signature heatmap and gene/metabolic network for 5 metabolic modules found in (**a**) displaying the strongest down-regulation (**b**) and up-regulation (**c**) upon the loss of both ErbB2 and ERRα signaling. Key significantly enriched pathways associated with DEGs within the metabolic networks are also shown. See also Supplementary Fig. 4.

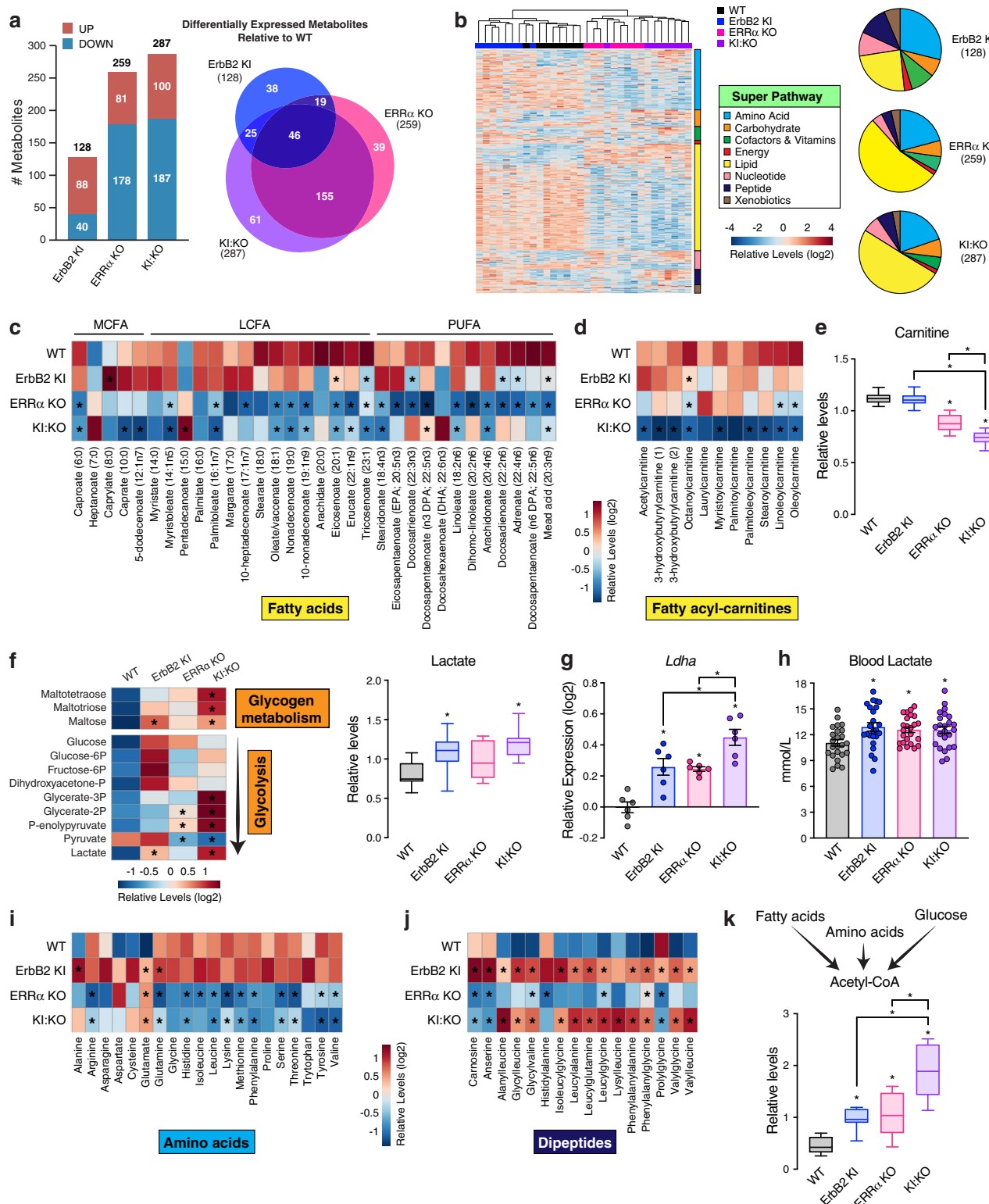

**Fig. 5 Characterization of ErbB2 KI, ERRα KO and KI:KO cardiac metabolomes. a** Bar chart and Venn diagram illustrating cardiac DEMs identified across genotypes relative to WT using a large-scale unbiased LC/MS-based screen (ANOVA, $p < 0.05$, $n = 8$). **b** Heatmap and pie charts of the cardiac mouse metabolomes subdivided into 8 major metabolic pathways. For the heatmap, DEMs were sorted from most up-regulated to most down-regulated in KI:KO hearts compared to WT within each metabolic category prior to unsupervised hierarchical sample clustering using Euclidean distance measure and average linkage. **c, d** Heatmap of cardiac fatty acid (**c**) and fatty acyl-carnitine (**d**) levels detected among the genotypes. **e** Box plot of free carnitine levels in mouse hearts ($n = 8$). **f** Heatmap of relative glycogen metabolite and glycolytic intermediate levels in the mouse hearts and box plot of heart lactate levels ($n = 8$). **g** RT-qPCR analysis of *Ldha* expression in mouse hearts ($n = 6$). Data are normalized to *Rplp0* levels. **h** Bar chart of circulating lactate levels in the mice ($n = 24$). **i, j** Heatmap of amino acid (**i**) and dipeptide (**j**) profiles in the mouse hearts. **k** Box plot of mouse cardiac acetyl-CoA levels ($n = 8$), produced by the breakdown and oxidation of lipids, carbohydrates, and protein. Data in (**e**, **f**, and **k**) are shown as box and whiskers plots: center line denotes median, box extends from 25th to 75th percentiles, and whiskers extend to the lowest and highest values; *$p < 0.05$ by ANOVA relative to WT controls, unless otherwise indicated. Data in (**g** and **h**) represent means ± SEM; *$p < 0.05$ by ANOVA relative to WT controls, unless otherwise indicated. Metabolites with an asterix (*) in the heatmap representations (**c**, **d**, **f**, **i**, **j**) were found significantly altered in the indicated mouse model versus WT, *$p < 0.05$ by ANOVA. See also Supplementary Fig. 5 and 6.

remodeling events including deregulation of phospholipid metabolism, up-regulation of glycolysis and sphingolipid metabolism, and down-regulation of FAO, pyruvate, and amino acid metabolism, extending the role of ERRα in metabolism to the control of lipid membrane homeostasis. Diminished pyruvate levels were attributed to increased PDH activity, a gatekeeper of glucose oxidation, due to decreased phosphorylation of the inhibitory site of Pdha1 at S232. Increased glycolytic rates in failing hearts have been conflictingly associated with either depressed or enhanced pyruvate oxidation[64,65]. Third, several commonalities between loss of ErbB2 and ERRα signaling were determined. Key effects included cardiomyocyte structural and contractile force protein modifications with links to perturbed GSK3, ERK, CaMKII, PKA, and PKC signaling. Mutations or variants of several co-targets serve as markers of human DCM and/or other cardiomyopathies such as desmoplakin (Dsp), filamin C (Flnc), myomesin 1 (Myom1), striated muscle enriched protein kinase (Speg), troponin I, cardiac muscle (Tnni3), and titin (Ttn)[66–68]. Loss of either factor contributed independently to acetyl-CoA and 2-HG accumulation, both biomarkers of failing hearts[69,70], with implications for epigenetic reprogramming events. While inhibition of either ErbB2 or ERRα signaling increased the reliance on a fetal metabolic program, a hallmark of declining heart health[71], their combined loss-of-function resulted in a sharp decrease in fatty acyl-carnitine species further increasing the dependence of the heart on glucose as cellular energy fuel. The observed up-regulation of AKT and ERK1/2 signaling as well as phosphorylation of Cofilin 1 at S3 upon dual inhibition of ErbB2 and ERRα signaling is an indication of profound cardiac remodeling and deregulated sarcomeric structure and function, underlying features of the aggravated DCM pathogenesis in KI:KO mice. Overall, altered structural integrity/mechanical stress and metabolic inflexibility are likely major contributing factors to increased reliance on glucose upon ErbB2 and ERRα loss, respectively. Finally, several mechanisms implicated in doxorubicin-induced cardiotoxicity were instilled by ERRα ablation including decreased FAO, increased glucose utilization, inflammation, and fibrosis, as well as perturbed calcium signaling[72]. Consistently, a compiled cardiac doxorubicin multi-omics signature was characteristic of decreased ERRα activity, harboring 89 molecules found similarly modulated by ERRα inhibition including key genes (eg *Acsl1*, *Hadh*, *Hmox1*, *Mmp2*, *Mycn*, *Nfe2l2*, *Pkia*) and metabolites (e.g. carnitine, arachidonate, myo-inositol). Doxorubicin is the most widely prescribed anthracycline, supporting the wide-ranging efforts for identification of the precise causal mechanisms underlying its cardiotoxicity. Whether impaired ERRα activity is indeed a contributor to doxorubicin-induced cardiotoxicity in the clinical setting remains to be validated. Intriguingly, statins, which harbor anti-oxidative and anti-inflammatory benefits, display ERRα agonism in vitro[73], and

their use has been shown to reduce the cardiotoxic risk of HER2 + patients treated with trastuzumab and/or anthracyclines such as doxorubicin[74,75].

This multi-omics study provides a rich resource of data supporting key roles for both ErbB2 and ERRα in the maintenance of normal cardiac function, however, we note several limitations. First, the phosphoproteomics and metabolomics studies involve MS-based identifications which will not capture all molecules present in each sample and can lead to false positives. Second, proteome-level changes were not evaluated and thus could not serve to normalize the phosphoproteome-level changes. Third, albeit the construction of integrated omics signatures filtered for increased potential for DCM causality, further investigation is needed to identify the precise causal pathogenic mechanisms instilled by ErbB2 and/or ERRα loss.

Our comprehensive investigation of cardiac regulomes provides key insights into the critical roles of ErbB2 and ERRα in the heart, characterizing integrated omics signatures of impaired ErbB2 and/or ERRα signaling linked to the pathogenesis of DCM, found most prominent upon their combined loss of action. In conclusion, the combinatory use of defined mouse models with integration of in-depth multi-omics analyses has proven to be a valuable approach to reveal several disturbed signaling pathways and thus potential cardiovascular risks in both prospective and current therapeutic strategies to treat cancer and metabolic diseases.

## Methods

**Animals**. All animal experimentations were conducted in accord with accepted standards of humane animal care and all protocols were approved by the McGill Facility Animal Care Committee and the Canadian Council on Animal Care. For all mouse experiments, male mice aged 15 weeks were sacrificed by cervical dislocation at Zeitgeber time (ZT) 4 for serum and tissue isolations. WT, ERRα KO[24], ErbB2 cDNA knock-in (ErbB2 KI)[25], and KI:KO mice generated from the cross of ErbB2 KI mice with ERRα KO mice in an FVB genetic background were housed at 22 °C under a 12-h light/dark cycle and fed ad libitum with free access to water in an animal facility at McGill University.

**Mouse Echocardiography**. Mice were lightly anaesthetized using 1-1.5% iso-flurane in oxygen and placed on a heated platform to maintain body temperature. Cardiac function and morphology were assessed by transthoracic echocardiography using a Vevo 2100 High-Resolution Imaging System with a 40 MHz MS 550D transducer (VisualSonics). Parasternal long-axis projection was used for orientation and left ventricular end-systolic and end-diastolic internal diameters were determined by two-dimensional M-mode images of a short-axis view at the proximal level of the papillary muscles. Ejection fraction and fractional shortening were calculated using VisualSonics Cardiac Measurements software included in the Vevo 2100 system following manual delineation of endocardial and epicardial borders in the parasternal short-axis cine loop.

**Histology**. Mouse hearts were cut on the short axis (transverse cut) and fixed in 10% buffered formalin for 48 h followed by paraffin embedding and serial sectioning (5 μm sections). For histological examination, slides were stained with Hematoxylin and Eosin (H&E) or Masson's trichrome (fibrosis marker). For

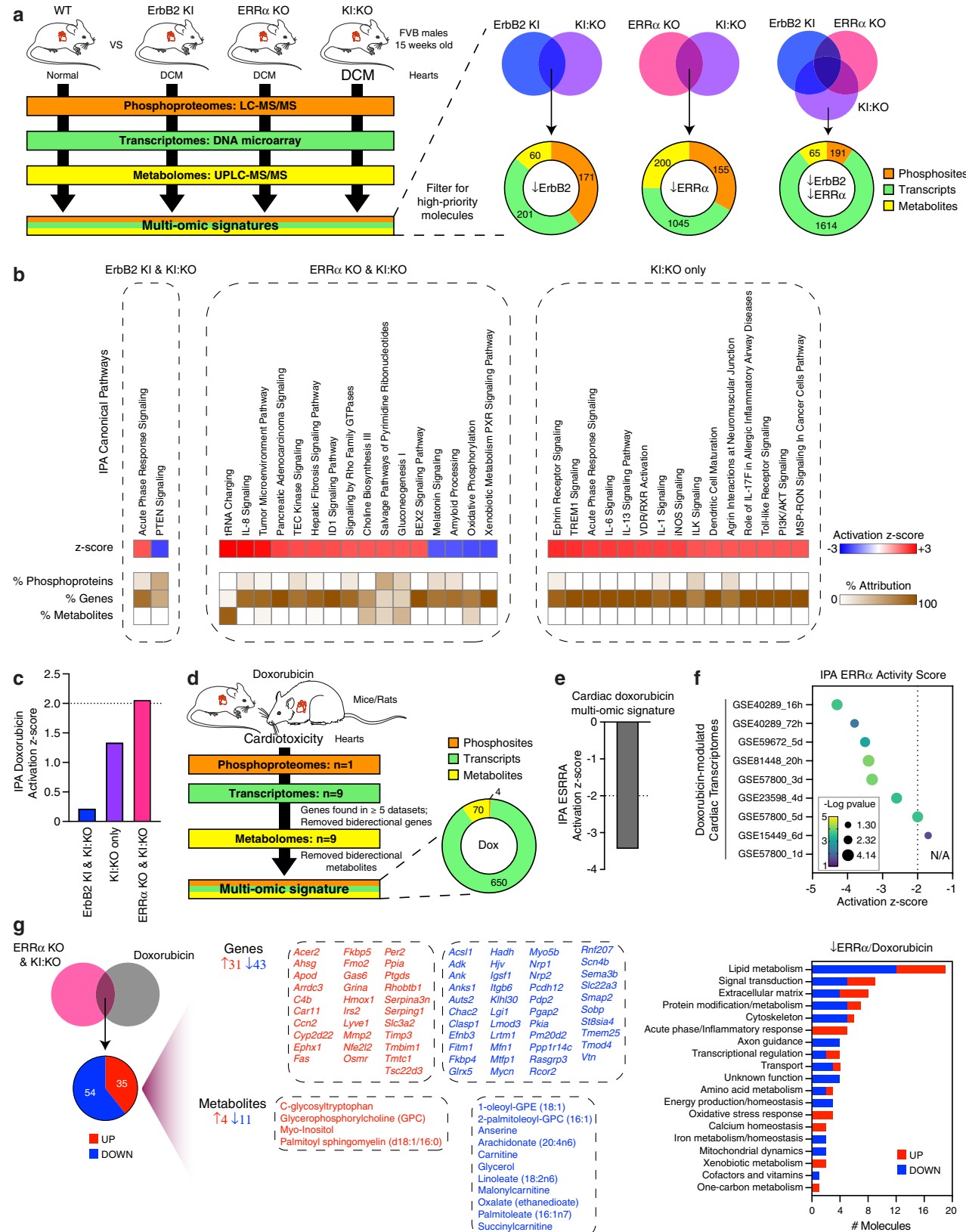

immunohistochemistry analysis, heart sections were immunostained with allograft inflammatory factor 1 (Aif-1, also known as Iba-1, macrophage infiltration marker, FUJIFIILM Wako Pure Chemical Corporation, Cat# 019-19741, RRID:AB_839504), CD31 (PECAM-1, endothelial marker for blood vessel density, Cell Signaling Technology, Cat# 77699, RRID:AB_2722705), or α-SMA (α-smooth muscle actin, Thermo Fisher Scientific, Cat# 14-9760-82, RRID:AB_2572996). Immunofluorescence staining with wheat germ agglutinin (WGA) conjugated to

Alexa Fluor 488 (Invitrogen, W11261) was performed to determine mean cardiomyocyte cross-sectional area. Apoptotic levels were evaluated using terminal deoxynucleotidyl transferase dUTP nick-end labeling (TUNEL) assay with the ApopTag Peroxidase In Situ Apoptosis Detection Kit (Millipore, S7100). Images of WGA-stained slides were taken with a Zeiss Axio Scan.Z1 instrument using a 20x objective and *Fiji* software[76] was used to quantify cardiomyocyte cross-sectional area from manual delineation of cell contours. Other slides were scanned with an

**Fig. 6 Characterization of cardiac ErbB2- and ERRα-dependent multi-omics signatures. a** Schematic illustration showing the integration of multi-omics signatures filtered for prioritizing deregulated DCM-causing molecules that are ErbB2- and/or ERRα-dependent. **b** IPA canonical pathway activity relationships with the 3 identified multi-omics signatures identified in (**a**). Pathways with significantly associated activation (z-score ≥ 2) or inhibition (z-score ≤ −2) states are shown with the relative contribution of each omics layer (phosphoprotein, gene, metabolite) to the predictions. **c** IPA identified doxorubicin as an upstream chemical drug with a significant activation score (z-score ≥ 2) uniquely in the ERRα-driven signature identified in (**a**). **d** Schematic illustration showing the integration of multi-omics signatures from doxorubicin-treated animals using publicly available phosphoproteomics[39], transcriptomics[40–45], and metabolomics[46–54] datasets. **e** IPA identified ESRRA (ERRα) as a transcriptional regulator with a significant down-regulated activation score (z-score ≤−2) in the doxorubicin-driven multi-omics signature identified in (**d**). **f** IPA attributed a significant down-regulated activation score (z-score ≤−2) to ESRRA (ERRα) in 7 of 9 doxorubicin-modulated transcriptomes used to construct the multi-omics signature in (**d**). **g** Intersection of the ERRα KO & KI:KO and doxorubicin cardiac multi-omics signatures identified in (**a, d**) resulted in 89 commonly deregulated molecules, mainly down-regulated and most largely associated with lipid metabolism. See also Supplementary Fig. 7.

Aperio ScanScope instrument (Aperio Technologies Inc.) and viewed with Aperio's ImageScope software. Quantification of interstitial fibrosis and Aif-1, CD31, and α-SMA immunoreactivity was performed using optimized Aperio digital analysis algorithms. TUNEL-positive cells were detected using HALO software (Indica Labs). Aside from WGA and TUNEL staining, histological procedures were performed by the Histology Core Facility of the Rosalind and Morris Goodman Cancer Institute (GCI).

**Biochemistry measurements and enzymatic assays**. Blood lactate was measured from the submandibular vein of mice using a Lactate Scout meter (Lactate.com). Collected blood was incubated at RT for 30 min prior to centrifugation at 5000 rpm for 30 min for serum separation. Serum free fatty acid levels were determined using a commercial quantification kit (Abcam, Cat# ab65341). Mouse cardiac pyruvate dehydrogenase (PDH) and pyruvate kinase (PK) enzymatic activities were measured using frozen tissue powder with commercial kits (Biovision, Cat# K679-100 and Abcam, Cat# ab83432, respectively).

**Mitochondrial content**. Cardiac mitochondrial (mt) content was assessed by measuring the ratio of mtDNA to nuclear DNA. DNA from frozen mouse heart tissue ($n = 6$/genotype) was isolated using a Qiagen DNeasy Blood & Tissue kit (Cat# 69504). mtDNA and nuclear DNA were amplified by qPCR using SYBR Green Master Mix (Roche) and a LightCycler 480 instrument (Roche) with specific primers for mitochondrial cytochrome b (*mt-Cytb*) and nuclear-encoded *Pgk2*, respectively, listed in Supplementary Data 6.

**Immunoblot analysis**. Whole-cell extracts from mouse hearts were prepared using Buffer K (sodium phosphate 20 mM, NaCl 150 mM, NP40 1%, EDTA 5 mM) containing protease and phosphatase inhibitor cocktail tablets (Roche) and quantified using the Bradford method (Bio-Rad Protein Assay, Cat# 5000006). Proteins (40 μg) were resolved on 8–12% SDS-PAGE gels then transferred onto PVDF membranes (Amersham Biosciences) and blocked for 1 h at RT in PBS-T (PBS + 0.1% Tween-20) containing 5% milk. Membranes were incubated overnight with primary antibodies (see below) diluted in PBS-T containing 5% milk. Following three washes in PBS-T + 5% milk, the membranes were incubated for 1 h with a secondary antibody either anti-rabbit (GE Healthcare, Cat# NA9340, Research Resource Identifier (RRID):AB_772191), anti-mouse (GE Healthcare, Cat# NA934, RRID:AB_772206), or anti-goat (Santa Cruz Biotechnology, Cat# sc-2020, RRID:AB_631728) diluted in PBS-T containing 5% milk. After washing three times with PBS-T, proteins were detected using ECL or ECL Prime Western Blotting detection reagent (Amersham). Primary antibodies used were: anti-phospho-Cx43 Ser368 (Cell Signaling Technology, Cat# 3511, RRID:AB_211016), anti-Cx43 (Millipore, Cat# 3512, RRID:AB_229459), anti-Vinculin (Santa Cruz Biotechnology, sc-25336, RRID:AB_628438), anti-phospho-Pdha1 Ser232 (Millipore, Cat# AP1063, RRID:AB_10616070), anti-Pdha1 (Cell Signaling Technology, Cat# 3205, RRID:AB_2162926), anti-Pdk1 (Enzo Life Sciences, Cat# ADI-KAP-PK112, RRID:AB_10618932), anti-Clock (Santa Cruz Biotechnology, sc-6927, RRID:AB_2082577), anti-α-SMA (Thermo Fisher Scientific, Cat# 14-9760-82, RRID:AB_2572996), anti-COX-2 (Cell Signaling Technology, Cat# 12282, RRID:AB_2571729), phospho-Myl2 Ser15 (Thermo Fisher Scientific, Cat# PA5-104265, RRID:AB_2816014), Myl2 (Cell Signaling Technology, Cat# 3672, RRID:AB_10692513), phospho-Tau Ser396 (Ser385 in mouse) (Abclonal, Cat# AP1028, RRID:AB_2863912), Tau (Abclonal, Cat# A0002, RRID:AB_2756869), Acadm (St John's Laboratory, Cat# STJ96389, RRID:AB_2922676), Acsl1 (Cell Signaling Technology, Cat# 4047, RRID:AB_2222411), Tfam (Abcam, Cat# ab131607, RRID:AB_11154967), phospho-AKT Ser473 (Cell Signaling Technology, Cat# 9271, RRID:AB_329825), phospho-AKT Thr308 (Cell Signaling Technology, Cat# 9275, RRID:AB_329828), AKT (Cell Signaling Technology, Cat# 9272, RRID:AB_329827), phospho-ERK1/2 (p44/42 MAPK) T202/Y204 (Cell Signaling Technology, Cat# 9101, RRID:AB_331646), ERK1/2 (p44/42 MAPK) (Cell Signaling Technology, Cat# 9102, RRID:AB_330744), phospho-Cfl1 S3 (Cell Signaling Technology, Cat# 3313, RRID:AB_2080597), and Cfl1 (Abclonal, Cat# A2658, RRID:AB_2922675). Immunoblot images were generated from film scanned with

an Epson Perfection V700 Photo scanner or obtained using a ChemiDoc MP imaging system (Bio-Rad, Cat# 12003154). Scanned images were cropped using Adobe Photoshop. Uncropped immunoblots are presented in Supplementary Fig. 8.

**Phosphoproteomics—protein lysis and digestion**. For each sample ($n = 5$ per group), 20 mg of frozen heart tissue powder (corresponding to an approximate protein amount of 1 mg) were lysed, reduced, and alkylated in lysis buffer (8 M Urea, 2 mM DTT, 100 uM Orthovanadate, supplemented with 40 mM iodoacetamide (IAA) after half an hour). The samples were then diluted in 50 mM ammonium bicarbonate to a Urea concentration of 1 M. Proteins were digested overnight at 25 °C with trypsin (Promega) with an enzyme/substrate ratio of 1:250. After digestion, samples were supplemented with 2 volumes of acetonitrile (ACN) and 6% trifluoroacetic acid (TFA) and precipitate was eliminated after a 10 min centrifugation in an Eppendorf centrifuge at 14,000 rpm.

**Phosphoproteomics—phosphorylated peptide enrichment**. The supernatant was transferred to Eppendorf tubes containing 10 μL of 5 μm TiO₂ beads (Canadian Biosciences) and incubated for 30 min on a rotating wheel. The supernatant was collected and the TiO2 beads were washed twice with 50% ACN/0.5% TFA in 200 mM NaCl and once with 50% ACN/0.1% formic acid (FA). Phosphopeptides were eluted with 10% ammonia in 50% ACN. This process was repeated once, to ensure quantitative phosphopeptide capture. Samples were dried down in a speed vac, resuspended in 20 μL of 0.1% FA in water and stored frozen, if not processed immediately.

**Phosphoproteomics — Mass Spectrometry: RP-nanoLC-MS/MS**. The data were acquired using an UHPLC Easy nLC 1000 (Thermo Scientific) coupled to an Orbitrap Q Exactive HF mass spectrometer (Thermo Scientific). 50% of the phosphopeptide enriched samples were first trapped (Acclaim PepMap 100 C18, 3 μm, 2 cm) before being separated on an analytical column (Acclaim, C18 2 μm, 25 cm). Trapping was performed in solvent A (0.1% FA in water), and the gradient was as follows: 2-20 % solvent B (0.1% FA in ACN) in 90 min, 20-38 % in 60 min, 38-90% in 10 min, maintained at 90% for 10 min, then, back to 0% solvent B in 10 min. The mass spectrometer was operated in data-dependent mode. Full-scan MS spectra from $m/z$ 375–1500 were acquired at a resolution of 120,000 at $m/z$ 400 after accumulation to a target value of $5 \times 10^6$. Up to 25 most intense precursor ions were selected for fragmentation. HCD fragmentation was performed at normalized collision energy of 35% after the accumulation to a target value of $1 \times 10^5$. MS/MS was acquired at a resolution of 30,000. A dynamic exclusion was set at 6 seconds.

**Phosphoproteomics—Mass spectrometry searches and data analysis**. Raw mass spectrometry data was searched using MaxQuant software (version 1.6.1.0) allowing for 3 missed trypsin cleavage sites, a fixed carbamidomethyl modification on cysteine (C) residues, and variable modifications on specified residues, including: oxidation (M), phosphorylation (S, T, Y), deamidation (N, Q), acetylation (protein N-terminus). First search peptide tolerance was set at 20 ppm and main search peptide tolerance was set at 4.5 ppm. Protein identification tolerance was set at a 1% false discovery calculated based on the search of a reverse sequence decoy database. Second peptide search and matching between run (0.7 min match time window, 20 min alignment time window) settings were enabled. Phosphorylation site search results (Phospho(STY)Sites.txt) were processed using Perseus (version 1.6.0.7). MaxQuant label-free intensities were used for quantification of phosphorylation site data (based on precursor ion intensity) after first filtering to remove identifications from the reverse database, phosphorylation sites with localization probability lower than 0.7, and those sites for which quantitative values were not found in at least 3 samples from at least one experimental group. The label-free intensities reflect the sum of all peptide intensities attributed to a specific protein group. There was no imputation of missing values and data were log2 transformed and normalized by width adjustment in Perseus. For this normalization, the first, second and third quartile (q1, q2, q3) are calculated from the

distribution of all values. The second quartile (which is the median) is subtracted from each value to center the distribution. Then we divide by the width in an asymmetric way. All values that are positive after subtraction of the median are divided by q3 - q2 while all negative values are divided by q2 - q1. Significant deregulated phosphopeptides between groups were determined using the limma[77] R/bioconductor package (p < 0.05) and 1.5 linear fold changes as the cut-offs. False discovery rates were estimated for each comparison using Benjamini–Hochberg adjusted p-values. Volcano plots and heatmap were generated using GraphPad Prism (version 9) and Morpheus (https://software.broadinstitute.org/morpheus/), respectively. For the latter, unsupervised hierarchical clustering was performed using Euclidean distance measure and average linkage. Principal component analysis (PCA) was performed using Phantasus version 1.11.0 (https://artyomovlab.wustl.edu/phantasus/) with the total list of 709 differentially expressed phosphopeptides as input identified across all 3 models relative to WT controls. Normalized quantitative values and group comparisons of phosphoproteomics data are summarized in Supplementary Data 1. Raw MS phosphoproteomics data have been deposited to the ProteomeXchange Consortium via the PRIDE[78] partner repository with the dataset identifier PXD032766.

Phosphopeptides significantly altered in the genetic mouse models relative to WT were searched for consensus motifs in 13-mer pSer and pThr sequences separately using PHOSIDA[79], iceLogo[80], and MoMo[27] algorithms after filtering to remove duplicates due to multiplicity effect and ambiguous identifications. For PHOSIDA, the De Novo Motif Finder tool was used with default parameters (minimum score = 15, which denotes a cutoff p-value $10^{-15}$, and minimum proportion of matching sites = 5%). For iceLogo, a mouse precompiled Swiss-Prot composition was used as the reference set with a percentage scoring method and cutoff p-value 0.05. For MoMo (version 5.4.1), the analysis was based on the Motif-x[28] algorithm using a shuffled peptide mix from the foreground sequences as the background, minimum occurrence threshold of 5%, and cutoff p-value $1^{-6}$. Kinases predicted to target the significantly enriched phosphomotifs were determined using the PhosphoMotif Finder tool[29] (http://www.hprd.org/PhosphoMotif_finder).

KEA2[30] (https://www.maayanlab.net/KEA2/) analysis of significant differentially phosphorylated phosphosites between groups served to identify enriched biological terms. As this software tool was limited to human phosphosite data, the altered mouse phosphosites were first mapped to human (Supplementary Data 1). IPA (Qiagen, Spring release version 2022) analysis using genes mapped to significant DEPPs between groups was used to identify enriched canonical pathways. GO Cellular Component (2018) enrichment analysis of genes mapped to significant DEPPs between groups was performed using the gene set search engine Enrichr[81] (https://maayanlab.cloud/Enrichr/).

**RNA isolation, reverse transcription and RT-qPCR**. Total RNA was extracted from mouse hearts using the RNeasy Fibrous Tissue Mini Kit (Qiagen, Cat# 74704). cDNA was made from 1 µg of RNA by reverse transcription with Random Primer Mix, dNTPs, 5X ProtoScript II RT Reaction buffer, DTT, RNAse inhibitor and ProtoScript II Reverse Transcriptase (NEB). cDNA was amplified by RT-qPCR using SYBR Green Master Mix (Roche) and a LightCycler 480 instrument (Roche) with specific primers listed in Supplementary Data 6. Relative expression was normalized to *Rplp0* levels.

**Microarray and transcriptome analysis**. Mouse heart microarray analyses were performed at the McGill University Génome Québec Innovation Centre. Samples were run on Affymetrix Genechip Mouse gene 2.0 ST arrays following Affymetrix's standard procedures (n = 3 per group). Microarray data have been deposited in the NCBI Gene Expression Omnibus (GEO) under accession number GSE199150. The data were analyzed using Expression Console (version 1.4.1) and Transcriptome Analysis Console (TAC, version 3.0) software (Affymetrix, Inc.). Significant DEGs between groups were determined by one-way ANOVA (p < 0.05) and 1.2 linear fold changes as the cut-offs (Supplementary Data 2). Unsupervised hierarchical clustering heatmap of DEGs using RMA expression values was generated using Morpheus with Euclidean distance measure and average linkage (https://software.broadinstitute.org/morpheus/). Matrisome genes, encoding ECM and ECM-associated proteins, were retrieved from the matrisome project[82] web platform (http://matrisomeproject.mit.edu) to generate heatmaps by Morpheus of matrisome DEGs using RMA expression values.

Functional enrichment analysis of DEGs was performed using Gene Set Enrichment Analysis (GSEA, https://www.gsea-msigdb.org/gsea/index.jsp) to identify enriched hallmark signatures within the Molecular Signature Database (MSigDB, version 5.2), IPA (Qiagen, Spring release version 2022) to identify enriched cardiac-related toxicological functions, and Enrichr[81] (https://maayanlab.cloud/Enrichr/) to identify enriched GO Cellular Components (2018).

The transcriptome-based metabolic network clustering analysis was done as described previously[34,35]. Briefly, similar to GAM metabolic network analysis[33], a metabolic network (graph) of reactions from KEGG database is considered. In the graph, the method tries to find a set of connected subgraphs (metabolic modules), with each corresponding well to a certain gene expression pattern. The initial patterns are defined using k-means clustering on gene expression matrix and then are refined in an iterative process using the network connections. Each derived metabolic module is presented as a graph that has vertices corresponding to

metabolites and the edges corresponding to the reactions with the expressed genes. Edge color represents the correlation score for a given DEG-encoding enzyme (green – lower correlation, red – higher correlation). Edge label size and width increase proportionally with the correlation score. Metabolic pathways associated with DEGs found within each metabolic module were determined by KEGG and REACTOME pathway enrichment analyses.

**Metabolomics—sample preparation**. Hearts were isolated from 15-week-old male WT, ErbB2 KI, ERRα KO, and KI:KO mice, washed in ice-cold PBS, and quickly flash-frozen in liquid nitrogen. Frozen whole mouse hearts (n = 8 per group) were submitted to Metabolon Inc. (Durham, NC, USA) for sample preparation and global cardiac metabolomics profiling analysis. Samples were prepared using the automated MicroLab STAR® system from Hamilton Company. Several recovery standards were added prior to the first step in the extraction process for QC purposes. To remove protein, dissociate small molecules bound to protein or trapped in the precipitated protein matrix, and to recover chemically diverse metabolites, proteins were precipitated with methanol under vigorous shaking for 2 min (Glen Mills GenoGrinder 2000) followed by centrifugation. The resulting extract was divided into five fractions: two for analysis by two separate reverse phase (RP)/ultrahigh performance (UP)LC-MS/MS methods with positive ion mode electrospray ionization (ESI), one for analysis by RP/UPLC-MS/MS with negative ion mode ESI, one for analysis by HILIC/UPLC-MS/MS with negative ion mode ESI, and one sample was reserved for backup. Samples were placed briefly on a TurboVap® (Zymark) to remove the organic solvent and stored overnight under nitrogen before preparation for analysis.

**Metabolomics—quality controls**. Several types of controls were analyzed in concert with the experimental samples: a pooled matrix sample generated by taking a small volume of each experimental sample (or alternatively, use of a pool of well-characterized human plasma) served as a technical replicate throughout the dataset; extracted water samples served as process blanks; and a cocktail of QC standards that were carefully chosen not to interfere with the measurement of endogenous compounds were spiked into every analyzed sample, allowed instrument performance monitoring and aided chromatographic alignment. Instrument variability was determined by calculating the median relative standard deviation (RSD) for the standards that were added to each sample prior to injection into the mass spectrometers. Overall process variability was determined by calculating the median RSD for all endogenous metabolites (i.e., non-instrument standards) present in 100% of the pooled matrix samples. Experimental samples were randomized across the platform run with QC samples spaced evenly among the injections.

**Metabolomics—UPLC-MS/MS**. All methods utilized a Waters ACQUITY ultra-performance liquid chromatography (UPLC) and a Thermo Scientific Q-Exactive high resolution/accurate mass spectrometer interfaced with a heated electrospray ionization (HESI-II) source and Orbitrap mass analyzer operated at 35,000 mass resolution. The sample extract was dried then reconstituted in solvents compatible to each of the four methods. Each reconstitution solvent contained a series of standards at fixed concentrations to ensure injection and chromatographic consistency. Method 1/Pos 1: One aliquot was analyzed using acidic positive ion conditions, chromatographically optimized for more hydrophilic compounds. In this method, the extract was gradient-eluted from a C18 column (Waters UPLC BEH C18-2.1×100 mm, 1.7 µm) using water and methanol, containing 0.05% perfluoropentanoic acid (PFPA) and 0.1% formic acid (FA). Method 2/Pos 2: A second aliquot was also analyzed using acidic positive ion conditions; however, it was chromatographically optimized for more hydrophobic compounds. In this method, the extract was gradient-eluted from the same aforementioned C18 column using methanol, acetonitrile, water, 0.05% PFPA and 0.01% FA, and was operated at an overall higher organic content. Method 3/Neg 1: A third aliquot was analyzed using basic negative ion optimized conditions using a separate dedicated C18 column. The basic extracts were gradient-eluted from the column using methanol and water, however with 6.5 mM Ammonium Bicarbonate at pH 8. Method 4/Neg 2: The fourth aliquot was analyzed via negative ionization following elution from a HILIC column (Waters UPLC BEH Amide 2.1 × 150 mm, 1.7 µm) using a gradient consisting of water and acetonitrile with 10 mM Ammonium Formate, pH 10.8. The MS analysis alternated between MS and data-dependent $MS^n$ scans using dynamic exclusion. The scan range varied slightly between methods but covered 70-1000 m/z.

**Metabolomics—data extraction and compound identification**. Raw data were extracted, peak-identified, and QC processed using Metabolon's hardware and software. These systems are built on a web-service platform utilizing Microsoft's .NET technologies, which run on high-performance application servers and fiber-channel storage arrays in clusters to provide active failover and load-balancing. Peaks were quantified using area-under-the-curve. Compounds were identified by comparison to library entries of purified standards or recurrent unknown entities. Metabolon maintains a library based on authenticated standards that contains the retention time/index (RI), mass to charge ratio (m/z), and chromatographic data (including MS/MS spectral data) on all molecules present in the library. Furthermore, biochemical identifications are based on three criteria:

retention index within a narrow RI window of the proposed identification, accurate mass match to the library ±10 ppm, and the MS/MS forward and reverse scores between the experimental data and authentic standards. The MS/MS scores are based on a comparison of the ions present in the experimental spectrum to the ions present in the library spectrum. While there may be similarities between these molecules based on one of these factors, the use of all three data points can be utilized to distinguish and differentiate biochemicals. More than 3300 commercially available purified standard compounds had been acquired and registered into a Laboratory Information Management System (LIMS) for analysis on all platforms for determination of their analytical characteristics. Additional mass spectral entries have been created for structurally unnamed biochemicals, which have been identified by virtue of their recurrent nature (both chromatographic and mass spectral). These compounds have the potential to be identified by future acquisition of a matching purified standard or by classical structural analysis.

A variety of curation procedures were carried out to ensure that a high-quality dataset was made available for statistical analysis and data interpretation. The QC and curation processes were designed to ensure accurate and consistent identification of true chemical entities, and to remove those representing system artifacts, mis-assignments, and background noise. Metabolon data analysts use proprietary visualization and interpretation software to confirm the consistency of peak identification among the various samples. Library matches for each compound were checked for each sample and corrected if necessary. Raw LC/MS metabolomics data have been deposited to the EMBL-EBI metabolomics repository, MetaboLights[83], with the dataset identifier MTBLS795.

**Metabolomics—data analysis.** For each of the 571 biochemicals identified with known identity across the mouse models, metabolite values expressed as raw area counts were rescaled to set the median equal to 1 and missing values were imputed with the minimum. Significant DEMs between groups were determined by one-way ANOVA contrasts ($p < 0.05$). False discovery rates were estimated for each comparison using the q-value method of Storey and Tibshirani[84]. Metabolomics data are summarized in Supplementary Data 3 with metabolites associated to 8 Super Pathways (amino acid, carbohydrate, cofactors/vitamins, energy, lipid, nucleotide, peptide, and xenobiotics).

Unsupervised hierarchical clustering heatmap generation using Euclidean distance and average linkage, supervised clustering using partial least squares-discriminant analysis (PLS-DA) and Random Forest classification of groups were performed with the statistical analysis module of MetaboAnalyst (version 5.0) using DEMs mapped (352 of 383) in the software tool. All other heatmaps of selected biochemical classes were generated with MetaboAnalyst.

**Integrated Omics Analysis.** Significant differentially expressed phosphosites, genes, and metabolites identified in ErbB2 KI, ERRα KO and KI:KO versus WT controls were first compiled (Supplementary Data 1–3). As loss of ErbB2 and ERRα independently contributed to the increased severity of DCM in KI:KO mice, biomolecules altered in ErbB2 KI and ERRα KO mice and commonly found modulated in KI:KO hearts were considered causal. Accordingly, integrated signatures from ErbB2 KI or ERRα KO hearts were filtered to contain only deregulated molecules commonly found in KI:KO hearts. Also, a third integrated signature found only in KI:KO hearts that necessitated the combined loss of both ErbB2 and ERRα was created. The ErbB2-, ERRα-, and ErbB2/ERRα-dependent integrated omics signatures are summarized in Supplementary Data 4. IPA (Qiagen, Spring release version 2022) analysis of these signatures was used to identify enriched and predicted activity states of canonical pathways (Supplementary Data 4) and upstream regulator analysis for doxorubicin activity prediction. Significant activity scores were considered as follows: activation, z-score ≥ 2; repression, z-score ≤ −2. Identifiers used for IPA analysis of the multi-omics signatures, DEGs identified by microarray analysis were mapped using Affymetrix IDs, DEMs identified by metabolomics were mapped using HMBD or CAS IDs, and DEPPs were mapped using UniProt IDs.

A cardiac doxorubicin multi-omics signature comprised of 724 molecules was created using available public datasets of differentially expressed phosphosites, genes, and metabolites limited to in vivo studies meeting specific inclusion criteria and summarized in Supplementary Data 5. Our search found 1 phosphoproteomics[39] investigation in doxorubicin-treated rats, however, due to technical limitations of the study, only 4 of 27 candidate altered phosphosites ($p < 0.05$) were included given their exact identification of the phosphorylated residue by MS or immunoblot validation. Transcriptomics studies with ≥ 500 identified DEGs ($p < 0.05$, |FC| ≥ 1.2) were considered. DEGs identified from 9 transcriptome datasets from 6 independent[40–45] studies on doxorubicin-treated animals (studies: rats $n = 4$; mice $n = 5$) were integrated together. DEGs with inconsistent directional changes within or across the datasets were removed and only filtered DEGs with consistent deregulations in 5 of 9 datasets were retained resulting in 650 genes. Metabolomics studies involving detection of ≥5 metabolites were considered. DEMs ($p < 0.05$) identified from 9 independent metabolomics[46–54] studies in doxorubicin-treated animals (studies: rats $n = 7$; mice $n = 2$) were integrated together. Non-specific metabolite determinations (mixtures) were disregarded and a total of 23 metabolites (Alanine, AMP, Arginine, Fumarate, Hexanolycarnitine, Histidine, Isoleucine, Lactate, Leucine, Lysine, Malate, N-delta-acetylornithine, Nicotinamide, Ornithine, Palmitate, Phenylalanine, Proline,

Serine, Serotonin, Stearate, Threonine, Tyrosine, and Valine) with inconsistent directional changes across the datasets were removed resulting in 70 DEMs. Noteworthy, the number of reliably perturbed metabolites by doxorubicin is likely higher given that most of the metabolomics studies employed targeted approaches. IPA (Qiagen, Spring release version 2022) analysis of the doxorubicin multi-omics signature was used to identify enriched and predicted activity states of canonical pathways (Supplementary Data 5) and upstream regulators (transcription factors/ nuclear receptors). Significant activity scores were considered as follows: activation, z-score ≥ 2; repression, z-score ≤ −2. Identifiers used for IPA analysis of the multi-omics signature, DEGs identified by mouse/rat microarray analyses were mapped using gene symbols, DEMs identified by metabolomics were mapped using HMBD or CAS IDs, and DEPPs were mapped using UniProt IDs.

**Statistics and reproducibility.** Data are presented as means ± SEM unless otherwise stated in the legends. For experimental group comparisons, statistical analyses were performed using limma or ANOVA with statistical significance defined as *$p < 0.05$. The number of biological replicates "$n$" define the number of individual mouse samples used per group/genotype for each experiment as indicted in the legends.

**Reporting summary.** Further information on research design is available in the Nature Research Reporting Summary linked to this article.

## Data availability

• Phosphoproteomics data have been deposited to the ProteomeXchange Consortium via the PRIDE[78] partner repository with the dataset identifier PXD032766. Microarray data have been deposited in the NCBI Gene Expression Omnibus (GEO) under accession number GSE199150. Metabolomics data have been deposited to the EMBL-EBI metabolomics repository, MetaboLights[83], with the dataset identifier MTBLS795. All omics datasets involving doxorubicin studies in mice or rats analyzed in the current study are publicly available and summarized in Supplementary Data 5.
• This paper does not report original code.
• Source data underlying the graphs are presented in Supplementary Data 7. Uncropped immunoblots are shown in Supplementary Fig. 8. Any additional information required to reanalyze the data reported in this paper is available from the lead contact upon request.

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

## Acknowledgements

The authors thank all members of the V.G. laboratory for helpful discussions, Gerardo Zapata from the Canadian Centre for Computational Genomics (C3G) for help with phosphoproteomics statistical analysis, and McGill GCI Research Support and Histology Core Facility. This work was supported by a Foundation Grant from the Canadian Institutes of Health Research (CIHR) (FRN-159933) and a Terry Fox Institute Program Project Grant to V.G. (PPG-1091). W.B. was supported in part by the McGill Integrated Cancer Research Training Program, H.X. is a recipient of a Fonds de Recherche du Québec – Santé (FRQS) Scholarship, and C.S is a recipient of a Canderel Scholarship.

## Author contributions
Conceptualization: C.R.D., M.-C.P., and V.G.; Study supervision: V.G.; Development of methodology: C.R.D., M-C.P., U.K., K.D., A.A.S.; Data acquisition: C.R.D., H.X., W.B., M-C.P., K.D., C.O., D.Z., V.S-G.; Data analysis: C.R.D., W.B., M-C.P., U.K., A.G., K.D., C.S., C.G., H.W.S., E.A-W., W.J.M., A.A.S., A.E., V.G.; Manuscript: C.R.D., M.-C.P., and V.G. with input from all authors.

## Competing interests
The authors declare no competing interests.
