## [Peer Review File · Communications Biology]

Reviewers' comments:

Reviewer #1 (Remarks to the Author):

Review of Dufour et al.

This manuscript describes the multi-omic characterization of three mouse models, that are relevant to the reported cardiotoxic impact of targeted breast cancer treatments.

Strengths: There are several. This manuscript presents a mountain of data including cardiac physiology, phosphoproteomics, transcriptomics, and metabolomics. The manuscript is organized and well-written. The data presentation is, overall, of very high quality. In fact, this is probably the best-presented multiomics study I have seen in quite a while. The manuscript makes a compelling case that care should be taken with combination breast cancer therapies, as the combined impact of ErBB2 and ERBB2 ablation on the heart is considerable in mice.

Weaknesses: In the past, the following issues have been sufficient to be deal-breakers for this reviewer, but the manuscripts were often also of lower quality overall. Nevertheless, I offer these points. Their weight, I leave to the editor's discretion. At a minimum, a "Limitations" in the discussion would be in order.

1. It looks like pairwise comparisons of phosphoproteomic data are analyzed by a 2-sample T-test. This is well-documented to be one of the least appropriate methods to analyze any omic data, as it lacks statistical power, increasing both false positives and false negatives. A Bayesian variance-pooling strategy including (but not limited to) LIMMA is preferred. Or, if the authors don't want to go Bayesian, it would still be an upgrade to perform ANOVA on all phosphopeptides then follow up with a pairwise contrast using something like Tukey's HSD.
2. For multigroup comparisons, ANOVA is used. However, for the number of replicates used, LIMMA generally outperforms ANOVA for both transcriptome and proteome studies.
3. The biggest limitation of the study, though, is that the phosphoproteome does not appear to be normalized for changes in the underlying expressed protein levels. This is non-trivial, since the authors showed in figure 1 that the mouse models differ with respect to myocyte CSA, as well as cardiac function. Therefore, it would be a safe assumption that there is all-around proteome remodeling. In fact, there is substantial remodeling of the transcriptome that would highly suggest changes in protein levels. The authors do validate specific examples where the phosphorylation status of the protein changes, while expression remains constant. However, they don't really know how often that holds true. Transcripts can't be used to normalize, as transcript and proteins changes in disease correlate only moderately. The bottom line is that for any given phosphosite changing say, 1.5-fold, the actual change at that site, when properly normalized to protein could be...well...anything.
4. The authors have gone with Metabolon for their metabolomics studies. The company has a reputation for providing one of the deeper metabolome platforms. However, unless their policies have changed, they do not provide the raw spectral data underlying the identification of specific metabolites. Therefore, the raw metabolomic data cannot be deposited in a metabolomic repository in accordance with prevailing guidelines for transparent and reproducible research. This has led some journals, including some journals of the American Chemical Society, to not accept manuscripts with data from Metabolon. I would urge both the authors and the editors to voice their concerns to the company.

Minor points

5. Page 5 "Cardiomyocytes were generally larger in hearts lacking ErbB2 (Fig. 1a,d)." Not sure you can say that when n=3 and not significant.

6. Figure 2. PCA or PLS-DA are important ways of summarizing all datasets. It would be nice to see a PCA plot for the phosphopeptide data.

7. Fig 2a.....Hmmm....n=224, n=231, n=410. I assume these are the number of phosphopeptides changing significantly not 224 biological replicates. The number of replicates is defined as n in the legend, so the authors should find an alternate way to call attention to the number of changing phosphopeptides.

8. Fig3a. Looks like the samples were hierarchically clustered using Euclidean distance and average linkage as stated in the methods, but it looks like the genes were likely sorted according to PC1 rather than agglomeratively-clustered. If so, this should also be reflected in the methods.

9. What peptide signals were quantified. IBAQ? LFQ?, MaxLFQ?

10. MS runs were logged, but how were they normalized or centered?

Reviewer #2 (Remarks to the Author):

The main claims of the paper are that the group found a large list of molecules and molecular mechanisms that explains the adverse cardiac remodeling in ErbB2 and ERRA deficiencies using a multi-omics approach.

However, the huge amount of information makes hard to identify the most meaningful findings which are not clearly highlighted by the authors until the discussion section. There is a big focus in signaling pathways that, in my experience, may be overestimated by IPA since there exist a huge overlapping elements between pathways, for instance, if the results display increased MAPK activity, several pathways would show upregulated by IPA without being necessarily true. Sometimes it is more meaningful to mention which molecules are altered rather than focus in the pathway which I encourage to mention in the result section. Also, sometimes there are some mechanisms that are overlooked by pathway analysis since there are only few reports on them, I encourage to give a mention on them (if any).

After doing a manual inspection of molecules that overlap in many pathways, potential upstream regulators could be spotted and a western blot could reveal which pathway was in fact activated.

The results provide a huge quantity of novel information that expands the knowledge in ErbB2 and ERRA that is relevant to understand potential mechanisms involved in cardiotoxicity secondary to breast cancer treatment. However, the study is highly focused in computational analysis and secondary validations of what the authors consider the most relevant findings are required to strength the findings.

Reviewer #3 (Remarks to the Author):

Re: Manuscript ID COMMSBIO-22-1061-T entitled, "Integrated multi-omics analysis of adverse cardiac remodeling and metabolic inflexibility upon ErbB2 and ERRA deficiency"

Summary: The authors utilize a phosphoproteomics, transcriptomics, and metabolomics to profile murine hearts deficient in either ErbB2, ERRA, or both ErbB2 and ERRA to profile molecular mechanisms associated with adverse cardiac remodeling in all three systems and their potential implications in contributing to cardiac dysfunction in the setting of anthracycline use.

Comments:

1. In the introduction, statements of incidence of cardiotoxicity in patients using herceptin being 3-7% and rising up to 27% should be clarified. In clinical studies, highest degree of cardiotoxicity is observed not solely in herceptin+doxorubicin, but is magnified in herceptin in combination with doxorubicin+cyclophosphamide (AC) therapy, AC is standard regimen in breast cancer treatment and should be clear that while the anthracycline alone is known to have magnified risk of cardiotoxicity solely, it is further worsened in subsequent adjuvant therapies. To this extent would provide brief summary of known mechanisms contributing to cardiotoxicity in the setting of these therapeutics and in discussion touch upon findings contributions to existing knowledge.
2. Considering the focus on the interaction between ErbB2 and ERRA in cardiac remodeling, the introduction would strongly benefit from discussing the known synergistic effects between ErbB2 and ERRA.
3. Criteria for DCM entails assessment of LV dimension and by echocardiography requires evaluating diastolic dimension- specifically LV internal cavity diameter (LVIDd) and secondary LV end diastolic diameter (LVEDd). It is unclear why LV end systolic (reported in text- graph not present in Fig 1) or LVIDs were selected. LVIDd should be reported. Additionally, the authors investigate cardiomyocyte hypertrophy, though hypertrophy is less likely observed in DCM.
 - a. A characteristic pathological feature of DCM is fibrosis which contributes to frequently seen diastolic dysfunction in DCM and cardiotoxicity induced by anthracyclines. Although understanding of why Trichrome data was included in Fig 3, its relevance to validating DCM pathology may be higher if moved to Figure 1 and discussed before WGA. Similarly, inflammation data may also be moved as a possible contributing factor but is of lower importance.
4. In Fig 1 B,C, F-H there are single * noted above some bars, but it is unclear what this significance is compared to. This is also observed in other figures; perhaps should define in legends * refers to comparison to WT and maybe use another symbol such as # to denote comparisons between 3 models.
5. Pg 5- ANP and BNP are markers of hemodynamic stress/heart failure- would specify as such rather than say "biomarkers of cardiovascular disease." Also in text Fig 1 I&J is referencing PCR data for ANP/BNP, however, graphs are just Fig 1J. Fig 1I is expression of ERRA and Errb2- would consider moving this earlier in Fig 1 or to supplemental figures to show validation of KI:KO model.
6. Was MoMo run for WT samples (Supplementary Figure 2 C,D)? If yes, should include for continuity of comparisons.
7. Pg 7 "IPA revealed DCM signaling in the top over-expressed pathways amongst the 3 groups with stronger perturbations in PKA, ILK,". In addition to DCM signaling, ILK is also common in all 3 models and should be clarified. For remaining pathways listed for the KI:KO model, would list in order seen in supplemental figure 3g to increase readability and understand the hierarchy of involvement.
8. It is interesting there is significant phosphorylation of Aldoa at S36 in KI:KO mice, which has been shown to be stimulated by PDGF, compared to remaining models. also known as fructose-bisphosphate aldolase involved in glycolysis and gluconeogenesis and a known key enzyme in metabolic reprogramming and metastasis in various cancer such as lung cancer.
9. For microarray analysis, an FC of ≥ 1.2 was selected. Why was FC of ≥ 1.5 not selected for a $p < 0.05$?
10. For Supplementary Figure S3b, would recommend listing terms in order of descending number of DEGs with priority for KI:KO model to more easily observed which cardiac targets may be more affected.
11. Pg 10, "...with 40% of top 15 distinguished metabolites annotated to lipid metabolism (eg acetyl-CoA, corticosterone) alongside metabolites to glycogen break down and glycolysis (pyruvate)." Should be state pyruvate is one of the top metabolites first (higher VIP score), though the majority of metabolites represent lipids.
12. Pg 13 "Intriguingly, a doxorubicin response..." prior to this statement, there was no mention of the purpose for doxorubicin treatment. A brief summary of purpose should be included to assist transition.
13. Regarding minor edits:

- Please fix spelling of “remodelling” to “remodeling” and “signalling” to “signaling” throughout text of the manuscript.
- Breast cancer (BCa) has already been defined on pg 3 of the manuscript; please ensure it is not repeated again and continue the use of the acronym for manuscript continuity (ex. Pg 4). Similarly, DCM has been identified earlier in introduction, on pg 7 use acronym vs saying “dilated cardiomyopathy.”
- Consider clarifying the background of KI:KO mice when discussing cross on pg 5 and also clarify the investigation of response in ERRa-KI and Erbb2 KO mice.
- Please consistently use acronyms such as ERRa-KI vs ERRa-null to increase readability.*
- Supplementary Fig 2E- if possible would edit “GAP” to say “Gja1” to be consistent with the text (pg 6). If incorrectly identified, please clarify the appropriate site of reference.
- Figure legends should clearly indicate which statistical test was employed.

Response to Reviewers

Reviewer 1:

Strengths: There are several. This manuscript presents a mountain of data including cardiac physiology, phosphoproteomics, transcriptomics, and metabolomics. The manuscript is organized and well-written. The data presentation is, overall, of very high quality. In fact, this is probably the best-presented multiomics study I have seen in quite a while. The manuscript makes a compelling case that care should be taken with combination breast cancer therapies, as the combined impact of Erca and ERB2 ablation on the heart is considerable in mice.

Weaknesses: In the past, the following issues have been sufficient to be deal-breakers for this reviewer, but the manuscripts were often also of lower quality overall. Nevertheless, I offer these points. Their weight, I leave to the editor's discretion. At a minimum, a "Limitations" in the discussion would be in order.

1. It looks like pairwise comparisons of phosphoproteomic data are analyzed by a 2-sample T-test. This is well-documented to be one of the least appropriate methods to analyze any omic data, as it lacks statistical power, increasing both false positives and false negatives. A Bayesian variance-pooling strategy including (but not limited to) LIMMA is preferred. Or, if the authors don't want to go Bayesian, it would still be an upgrade to perform ANOVA on all phosphopeptides then follow up with a pairwise contrast using something like Tukey's HSD.

Response:

We completely agree with the reviewer and appreciate this was pointed out. We have re-analyzed the phosphoproteomics data using limma statistical testing and updated the manuscript accordingly including methods, Supplemental Tables 1 and 4, and associated Figures 2, 6 and Extended Data Fig. 2, 3d, 7a.

2. For multigroup comparisons, ANOVA is used. However, for the number of replicates used, LIMMA generally outperforms ANOVA for both transcriptome and proteome studies.

Response:

As mentioned in point 1 above, we have now used limma for multigroup comparisons of the phosphoproteomics data. Regarding our transcriptome (microarray) study, ANOVA was used for identification of DEGs between the groups. We have not re-analyzed the transcriptome data using limma but will consider this for future work.

3. The biggest limitation of the study, though, is that the phosphoproteome does not appear to be normalized for changes in the underlying expressed protein levels. This is non-trivial, since the authors showed in figure 1 that the mouse models differ with respect to myocyte CSA, as well as cardiac function. Therefore, it would be a safe assumption that there is all-around proteome

remodeling. In fact, there is substantial remodeling of the transcriptome that would highly suggest changes in protein levels. The authors do validate specific examples where the phosphorylation status of the protein changes, while expression remains constant. However, they don't really know how often that holds true. Transcripts can't be used to normalize, as transcript and proteins changes in disease correlate only moderately. The bottom line is that for any given phosphosite changing say, 1.5-fold, the actual change at that site, when properly normalized to protein could be...well...anything.

Response:

We acknowledge that the phosphoproteome not being normalized to the proteome makes it virtually impossible to decipher whether the observed protein phosphorylation changes reflect phosphosite-level or protein-level changes or a combination of both globally. We now include mention of this limitation in the discussion as shown below.

“This multi-omics study provides a rich resource of data supporting key roles for both ErbB2 and ERR α in the maintenance of normal cardiac function, however, we note several limitations. First, the phosphoproteomics and metabolomics studies involve MS-based identifications which will not capture all molecules present in each sample, can lead to false positives, and the use of label-free quantification is semi-quantitative. Second, proteome-level changes were not evaluated and thus could not serve to normalize the phosphoproteome-level changes. Third, albeit the construction of integrated omics signatures filtered for increased potential for DCM causality, further investigation is needed to identify the precise causal pathogenic mechanisms instilled by ErbB2 and/or ERR α loss.”

Proteome and phosphoproteome analyses are generally not measured simultaneously on the same sample preparation as the latter involves special treatment to first enrich for phosphopeptides prior to detection given their lower abundance. This difference in sample preparation by itself can introduce error during phosphoproteome normalization to the proteome. In our case, where the proteome was not assessed, normalization methods do exist but nonetheless it is still acceptable to not normalize the phosphoproteome data as the normalization methods could by themselves also shift the observed changes in a wrong manner. Initially, we did not perform a normalization step on the presented phosphoproteomics data, however, we have carefully re-considered this and have proceeded with a normalization step (width adjustment in Perseus) to help ensure the samples were comparable prior to limma statistical testing. For this normalization, the first, second and third quartile (q1, q2, q3) are calculated from the distribution of all values. The second quartile (which is the median) is subtracted from each value to center the distribution. Then we divide by the width in an asymmetric way. All values that are positive after subtraction of the median are divided by q3 - q2 while all negative values are divided by q2 - q1.

Altered protein phosphorylation status is ultimately an indicator of altered protein function/activity. While we do not know the precise mechanism for all observed phosphorylation changes, the fact remains that the activities of the proteins identified are likely modulated. Thus, our analysis of the phosphoproteomics data on a protein-level should not be impacted by the

underlying mechanism. However, we acknowledge that phosphopeptide-level analyses sought to infer potential upstream kinases from over-represented phosphomotifs may be confounded by the potential contribution of underlying protein-level changes. In response to comments by Reviewer 2, we have included additional validation for observed or inferred deregulations within the groups, including the altered phosphorylation of Cx43 (Connexin 43), Myl2 (myosin light chain 2), Tau, and Cfl1 (Cofilin-1), in line with the phosphoproteomics results (Fig. 2g, Extended Data Fig. 7b,e).

Fig. 2g

Extended Data Fig. 7b

Extended Data Fig. 7e

4. The authors have gone with Metabolon for their metabolomics studies. The company has a reputation for providing one of the deeper metabolome platforms. However, unless their policies have changed, they do not provide the raw spectral data underlying the identification of specific metabolites. Therefore, the raw metabolomic data cannot be deposited in a metabolomic repository in accordance with prevailing guidelines for transparent and reproducible research. This has led some journals, including some journals of the American Chemical Society, to not accept manuscripts with data from Metabolon. I would urge both the authors and the editors to voice their concerns to the company.

Response:

We have reached out to Metabolon concerning this request. The company was able to offer upload of raw LC/MS data for the mass channel (no MS/MS data) to the MetaboLights repository for metabolomics studies upon a fee that we have accepted. We have now included the phrase, “Raw LC/MS metabolomics data have been deposited to the EMBL-EBI metabolomics repository,

MetaboLights⁸³, with the dataset identifier MTBLS795 (<https://www.ebi.ac.uk/metabolights/MTBLS795>).”

Minor point 1. Page 5 “Cardiomyocytes were generally larger in hearts lacking ErbB2 (Fig. 1a,d).” Not sure you can say that when n=3 and not significant.

Response:

Together with comments by Reviewer 3, we have remodeled Figure 1 and moved the cardiomyocyte size assessment data to Extended Data Fig. 1e. The phrase in question was removed and replaced by, “KI:KO displayed a synergistic effect of impaired ErbB2 and ERR α signaling on DCM development (Fig. 1e-i), not associated with greater vascular defects, cardiomyocyte apoptosis or hypertrophy (Fig. 1a and Extended Data Fig. 1b-e).”

Minor point 2. Figure 2. PCA or PLS-DA are important ways of summarizing all datasets. It would be nice to see a PCA plot for the phosphopeptide data.

Response:

We now include a PCA plot of the re-analyzed and normalized phosphoproteomics data (See response to major points 1 and 3 for the re-analysis). Using the total list of 709 differentially expressed phosphopeptides as input identified across all 3 models relative to WT controls (limma: $p < 0.05$, $|FC| \geq 1.5$), we clearly see biological replicate clustering within groups with ErbB2 KI samples displaying greater similarity to KI:KO samples (new Extended data Fig. 2c). The PCA plot concurs with the unsupervised hierarchical sample clustering heatmap (new Extended data Fig. 2b).

Figure S2

Minor point 3. Fig 2a.....Hmmm....n=224, n=231, n=410. I assume these are the number of phosphopeptides changing significantly not 224 biological replicates. The number of replicates is defined as n in the legend, so the authors should find an alternate way to call attention to the number of changing phosphopeptides.

Response:

We have updated Fig. 2a with the re-analyzed phosphoproteomics data and have made the # of up- and down-regulated phosphopeptides clearer with distinction from the use of “n”.

Figure 2

Minor point 4. Fig3a. Looks like the samples were hierarchically clustered using Euclidean distance and average linkage as stated in the methods, but it looks like the genes were likely sorted according to PC1 rather than agglomeratively-clustered. If so, this should also be reflected in the methods.

Response:

For the heatmap in Fig. 3a, DEGs were first sorted (manually) from most up-regulated to most down-regulated in KI:KO hearts compared to WT prior to unsupervised sample clustering using Euclidean distance measure and average linkage. The legend now specifies this.

Actually, the heatmap for differentially expressed phosphopeptides was generated similarly (new Extended data Fig. 2b). The same holds true for the heatmap of DEMs, except the sorting (from up- to down-regulated) in the KI:KO vs WT comparison was done for each of the 8 metabolic categories separately (Fig. 5b). We have now updated the legends to note this.

Minor point 5. What peptide signals were quantified. IBAQ? LFQ?, MaxLFQ?

Response:

In the manuscript, we now specify that, “MaxQuant label-free intensities were used for quantification of phosphorylation site data (based on precursor ion intensity)...”. We now also clarify that, “The label-free intensities reflect the sum of all peptide intensities attributed to a specific protein group”. In MaxQuant, this is defined as, “Summed up eXtracted Ion Current (XIC) of all isotopic clusters associated with the identified AA sequence”.

Minor point 6. MS runs were logged, but how were they normalized or centered?

Response:

As explained in Major point 3, our phosphoproteomics data was initially non-normalized. We have re-considered this and have proceeded with a normalization step (width adjustment in Perseus) prior to limma statistical testing. Again, for this normalization, the first, second and third quartile (q_1 , q_2 , q_3) are calculated from the distribution of all values. The second quartile (which is the median) is subtracted from each value to center the distribution. Then we divide by the width in an asymmetric way. All values that are positive after subtraction of the median are divided by $q_3 - q_2$ while all negative values are divided by $q_2 - q_1$.

Reviewer 2:

The huge amount of information makes hard to identify the most meaningful findings which are not clearly highlighted by the authors until the discussion section. There is a big focus in signaling pathways that, in my experience, may be overestimated by IPA since there exist a huge overlapping elements between pathways, for instance, if the results display increased MAPK activity, several pathways would show upregulated by IPA without being necessarily true. Sometimes it is more meaningful to mention which molecules are altered rather than focus in the pathway which I encourage to mention in the result section. Also, sometimes there are some mechanisms that are overlooked by pathway analysis since there are only few reports on them, I encourage to give a mention on them (if any).

After doing a manual inspection of molecules that overlap in many pathways, potential upstream regulators could be spotted and a western blot could reveal which pathway was in fact activated.

The results provide a huge quantity of novel information that expands the knowledge in ErbB2 and ERRA that is relevant to understand potential mechanisms involved in cardiotoxicity secondary to breast cancer treatment. However, the study is highly focused in computational analysis and secondary validations of what the authors consider the most relevant findings are required to strength the findings.

Response:

Indeed, the vast amount of data collected from the use of three omics platforms in this study makes the computational analysis challenging but essential. Redundancy of molecules across a multitude of pathways adds to the complexity in the interpretation of functional enrichment analyses. On the other hand, missing identifications from any omics layer as well as missing annotations or functional roles are also an issue. Nevertheless, functional tools help provide insight into otherwise complex data.

It is worth noting that based on the comments from Reviewer 1, the phosphoproteomics dataset was entirely re-analyzed with analyses/figures updated accordingly. A normalization step was added as well as the use of limma as the statistical testing method. The IPA software is periodically updated with expansion of their knowledgebase. As such, a newer version of the software was used. At the same time, we took the liberty to update our metabolomics analysis with the current version of MetaboAnalyst (version 5 from version 3). By doing so, 50 additional metabolites were mapped by the software as compared to previously for a total of 352 of 383 DEMs mapped and used in the analyses by MetaboAnalyst (heatmap, PLS-DA, Random Forest classification). Together, the phosphoproteomics re-analysis and updates in software tools and annotations have led to slight changes in the results from the original version (all methods were updated).

We agree with the reviewer that further validation would be beneficial to support key findings keeping in mind that a noted issue with phosphoproteomics validation is the limitation of available phospho-site specific antibodies.

Notable changes include:

- Western blot and IHC detection of α -SMA (α -smooth muscle actin) to further support the fibrogenic phenotypes in hearts lacking $ERR\alpha$ (Fig. 3f,g).
- Western blot detection of S368 phosphorylated Cx43 (Connexin 43) to support the reduced phosphorylated levels identified by phosphoproteomics analysis in hearts lacking ErbB2 (Fig. 2g).
- Western blot detection of COX-2 (cyclooxygenase-2) to support the transcriptional up-regulation of prostaglandin synthesis (Extended Data Fig. 6c).
- Western blot detection of Ser15 phosphorylated Myl2 (myosin light chain 2) to support the phosphoproteomics data, establishing dephosphorylation and inactivation of Myl2 in hearts lacking ErbB2, a likely contributor to DCM (Extended Data Fig. 7b).
- Western blot detection of Ser385 phosphorylated Tau to support the phosphoproteomics data, showing reduced levels in hearts lacking ErbB2 (Extended Data Fig. 7b).
- Western blot detection of *Acadm* and *Acs11*, both important for mitochondrial fatty acid oxidation, in support of their transcriptional down-regulation in hearts lacking $ERR\alpha$ (Extended Data Fig. 7c). These findings were further tied to decreased levels of *Tfam*, important for mitochondrial transcription and replication, in parallel with decreased mitoDNA:nuclearDNA content (Extended Data Fig. 7d).
- Western blot assessment of AKT and ERK1/2 phosphorylation status to support the IPA pathway activity/enrichment findings. Increased activities of both AKT and ERK1/2 were determined in KI:KO mice, supporting the IPA predicted up-regulation of PI3K/AKT activity and significant enrichment of altered molecules associated with ERK/MAPK signaling (Extended Data Fig. 7e).
- Western blot detection of Ser3 phosphorylated Cfl1 (Cofilin-1) to support the phosphoproteomics data, establishing hyperphosphorylation of Cfl1 S3 in KI:KO hearts, an indication of impaired sarcomeric actin dynamics (Extended Data Fig. 7e).

Fig. 3f

Fig. 3g

Fig. 2g

Extended Data. Fig. 6c

Extended Data Fig. 7b-e

Reviewer 3:

1. In the introduction, statements of incidence of cardiotoxicity in patients using herceptin being 3-7% and rising up to 27% should be clarified. In clinical studies, highest degree of cardiotoxicity is observed not solely in herceptin+doxorubicin, but is magnified in herceptin in combination with doxorubicin+cyclophosphamide (AC) therapy, AC is standard regimen in breast cancer treatment and should be clear that while the anthracycline alone is known to have magnified risk of cardiotoxicity solely, it is further worsened in subsequent adjuvant therapies. To this extent would provide brief summary of known mechanisms contributing to cardiotoxicity in the setting of these therapeutics and in discussion touch upon findings contributions to existing knowledge.

Response:

We have modified the introduction to help clarify the cardiotoxic risk statement in relation to standard treatment regimens and included mention of underlying potential contributing mechanisms although the precise causal mechanisms remain elusive as described below.

“Treatment regimens for HER2+ BCa typically involve ErbB2-targeted therapies including trastuzumab in combination with chemotherapies such as the anthracycline doxorubicin and alkylating agent cyclophosphamide to enhance the anti-tumor effects of HER2-blockade, albeit increasing the cardiotoxic risk from 3-7% to 27% of patients⁷⁻⁹. Impaired stress responses and cardiomyocyte apoptosis consequential to compromised cell survival and repair are implicated in trastuzumab-induced cardiac dysfunction¹⁰. Doxorubicin-induced adverse cardiac effects are the most severe, however both doxorubicin and cyclophosphamide can cause mitochondrial damage and dysfunction, oxidative and nitrate stress, calcium deregulation, inflammation, and fibrosis^{10,11}.”

In the discussion, we have modified the text to clarify or expand upon certain key points. We made it clearer that loss of ERR α activity in conjunction with ErbB2 signaling down-regulation increased the severity of DCM, a key finding of this study. Also, by including the following phrase, “To the best of our knowledge, no prior investigation globally mapped the ERR α -dependent phosphoproteome or metabolome in any cellular context and no prior study performed phosphoproteomics or metabolomics profiling downstream loss of ErbB2 signaling in the adult heart”, we emphasize the extent to which our findings shed new light on the cardioprotective roles of both ERR α and ErbB2. We further dive into the potential discovery that impaired ERR α activity may be a contributor to doxorubicin-induced cardiotoxicity in the clinic based on the finding of common molecular features ensuing ERR α loss-of-function and doxorubicin action in the heart. We have included the following statements to this regard, “Doxorubicin is the most widely prescribed anthracycline, supporting the wide-ranging efforts for identification of the precise causal mechanisms underlying its cardiotoxicity. Whether impaired ERR α activity is indeed a contributor to doxorubicin-induced cardiotoxicity in the clinical setting remains to be validated. Intriguingly, statins, which harbor anti-oxidative and anti-inflammatory benefits, display ERR α agonism *in vitro*⁷³, and their use has been shown to reduce the cardiotoxic risk of HER2+ patients treated with trastuzumab and/or anthracyclines such as doxorubicin^{74,75}.”. Further, we underscore key findings of the study supporting the cardioprotective roles of ErbB2

and $ERR\alpha$ in the heart. We emphasize the importance of ErbB2 signaling for cardiomyocyte architectural integrity via sarcomeric Myl2 and gap junction Cx43 regulation and $ERR\alpha$ in the control of cardiac metabolic programs and inflammation/fibrosis. The adverse cardiac remodeling observed in mice with concomitant loss of both ErbB2 and $ERR\alpha$ signaling, exhibiting worsened DCM pathogenesis, was linked to up-regulated cardiac AKT and ERK1/2 signaling as well as increased phosphorylation and deactivation of Cofilin-1 (Ser3), an essential regulator of actin turnover.

2. Considering the focus on the interaction between ErbB2 and $ERR\alpha$ in cardiac remodeling, the introduction would strongly benefit from discussing the known synergistic effects between ErbB2 and $ERR\alpha$.

Response:

We have modified the introduction accordingly as follows, “In malignant BCa, $ERR\alpha$ and ErbB2 are functionally linked and their expression levels correlate positively²⁰⁻²². Notably, attenuation of ERBB2 signaling disrupts $ERR\alpha$ activity²², and reciprocally, $ERR\alpha$ ablation reduces ERBB2 amplicon gene transcription and impedes ErbB2-induced murine BCa development²⁰.”

3. Criteria for DCM entails assessment of LV dimension and by echocardiography requires evaluating diastolic dimension- specifically LV internal cavity diameter (LVIDd) and secondary LV end diastolic diameter (LVEDd). It is unclear why LV end systolic (reported in text- graph not present in Fig 1) or LVIDs were selected. LVIDd should be reported. Additionally, the authors investigate cardiomyocyte hypertrophy, though hypertrophy is less likely observed in DCM.

A characteristic pathological feature of DCM is fibrosis which contributes to frequently seen diastolic dysfunction in DCM and cardiotoxicity induced by anthracyclines. Although understanding of why Trichrome data was included in Fig 3, its relevance to validating DCM pathology may be higher if moved to Figure 1 and discussed before WGA. Similarly, inflammation data may also be moved as a possible contributing factor but is of lower importance.

Response:

In Fig. 1h, the data presented reflects LVIDs = left ventricular internal diameter end systole. We have updated Figure 1 to include both LVIDs and LVIDd = left ventricular internal diameter end diastole (new Fig. 1h,i). We have clarified the definitions for LVIDs and LVIDd in the manuscript. The results support the development of DCM in our models.

Fig. 1h,i

Indeed, cardiomyocyte hypertrophy is not a hallmark of DCM. However, given the generation of a new mouse model with combined loss of function of ErbB2 and ERR α , we extended our analysis to include the assessment of myocyte size to be more comprehensive. Taking into consideration your other point to emphasize more on the fibrogenic phenotypes early on which is a strong underlying cause of DCM and considering also the point raised by Reviewer 1 that the significance of the myocyte size assessment was based on a limited sample size, we have pushed the Trichrome data to Fig. 1 and moved the cardiomyocyte assessment data to Extended Data Fig. 1. The inflammation data presented in Fig. 3 was maintained in this figure to support the transcriptome findings.

4. In Fig 1 B,C, F-H there are single * noted above some bars, but it is unclear what this significance is compared to. This is also observed in other figures; perhaps should define in legends * refers to comparison to WT and maybe use another symbol such as # to denote comparisons between 3 models.

Response:

In the legends, we specify that the use of an asterix “*” is to signify statistical significance of $p < 0.05$. When comparing the 3 genotypes to WT, we did not use a line to indicate these comparisons to avoid overcrowding the panels with lines. We have updated all the legends to clarify the statistical comparisons as for example, “* $p < 0.05$ by ANOVA relative to WT controls, unless otherwise indicated.”

5. Pg 5- ANP and BNP are markers of hemodynamic stress/heart failure- would specify as such rather than say “biomarkers of cardiovascular disease.” Also in text Fig 1 I&J is referencing PCR data for ANP/BNP, however, graphs are just Fig 1J. Fig 1I is expression of ERR α and Errb2- would consider moving this earlier in Fig 1 or to supplemental figures to show validation of KI:KO model.

Response:

We have modified the text regarding ANP and BNP as the reviewer pointed out along with reference to the appropriate figure panel for the related expression data.

“Consistent with their increased disease severity, KI:KO hearts expressed higher transcript levels of two biomarkers of hemodynamic stress and heart failure, atrial natriuretic peptide (ANP) and

brain natriuretic peptide (BNP)²⁶ (Fig. 1j), thus supporting both ERR α - and ErbB2-dependent contributions to the observed DCM.”

Also as suggested, expression data for ERR α and ErbB2 was moved to Extended Data Fig. 1a to support the concomitant loss of both factors in KI:KO mice.

6. *Was MoMo run for WT samples (Supplementary Figure 2 C,D)? If yes, should include for continuity of comparisons.*

Response:

Actually, all algorithms used for phosphomotif interrogation were based on the list of differentially expressed phosphopeptide sequences identified in each of the 3 genetically altered models (ErbB2 KI, ERR α KO, and KI:KO) relative to WT. Thus, there is no analysis in the WT group as the goal here was to help pinpoint potential kinases responsible for the phosphoproteome perturbations found in the 3 groups compared to normal WT controls. We clarified this in the manuscript (results/legends/methods).

7. *Pg 7 “IPA revealed DCM signaling in the top over-expressed pathways amongst the 3 groups with stronger perturbations in PKA, ILK, ...”. In addition to DCM signaling, ILK is also common in all 3 models and should be clarified. For remaining pathways listed for the KI:KO model, would list in order seen in supplemental figure 3g to increase readability and understand the hierarchy of involvement.*

Response:

In response to points raised by Reviewer 1 concerning the analysis of the phosphoproteome data, this dataset was completely re-analyzed, and all associated downstream analyses/figures were redone and updated accordingly. A normalization step was introduced as well as the use of limma as the statistical testing method used for multi-group comparisons (detailed in the methods section). New analyses with IPA were performed with an updated software version. We were more careful regarding the description of the new results of IPA over-represented pathways. To facilitate the interpretation of the presented data, we have included a heatmap representation of the associated number of DEPPs with the pathways identified but retained the order of pathways in descending order of p-value significance found in the KI:KO vs WT comparison (new Extended Data Fig. 2h). We have also applied this format for data presentation in Extended Data Fig. 2g and Extended Data Fig. 3c.

Extended Data Fig. 2g,h

Extended Data Fig. 3c

8. It is interesting there is significant phosphorylation of Aldoa at S36 in KI:KO mice, which has been shown to be stimulated by PDGF, compared to remaining models. also known as fructose-bis phosphate aldolase involved in glycolysis and gluconeogenesis and a known key enzyme in metabolic reprogramming and metastasis in various cancer such as lung cancer.

Response:

We appreciate the reviewer pointing out p-Aldoa Ser36 as a potential underlying contributor to the DCM severity in KI:KO, now found also significantly perturbed in ERR α KO following re-analysis of the phosphoproteomics data. Unfortunately, there are no currently available phospho-specific antibodies targeting Aldoa S36 for validation purposes. Nevertheless, we have now included mention of this modification in the manuscript.

“Phosphorylation of Aldoa at S36, found significantly elevated in ERR α KO and more prominently in KI:KO hearts, was recently found to drive glycolytic metabolism of liver cancer cells³¹.

9. For microarray analysis, an FC of ≥ 1.2 was selected. Why was FC of ≥ 1.5 not selected for a $p < 0.05$?

Response:

First, we can significantly validate a 20% change in gene expression by RT-qPCR. More importantly, ERR α is a master transcriptional regulator of metabolism, a core area of research in our laboratory, and its altered activity has been shown in multiple reports to have profound effects on metabolism, particularly mitochondrial energy metabolism (e.g., FAO, OXPHOS). In this manuscript, we show that ERR α loss greatly perturbs cardiac metabolism on a global level by metabolomics (Fig. 5), particularly lipid metabolism, found strongly supported by gene-level changes (Fig. 4). It is widely accepted that transcriptional changes of individual metabolic genes are small. However, a collection of small changes in multiple genes within a given metabolic pathway is highly significant as we found in Fig. 4 with deregulated metabolic modules. Given our expertise with metabolism and these points in mind, we set our microarray FC cut-off to 1.2. Use of a more stringent transcriptome FC cut-off of 1.5 would undoubtedly lose the majority if not all perturbed metabolic pathways found enriched or deregulated in the absence of ERR α . For instance, FAO was found as the most enriched pathway by IPA in the integrated omics signature identified in ERR α KO hearts found sustained in KI:KO hearts (Extended Data Fig. 7a). Use of a $|FC| \geq 1.5$ would result in loss of 75% of the contributing genes including *Acadm* and *Acs11* to the significance. Reduced levels of *Acadm* and *Acs11* protein were confirmed and now shown in Extended Data Fig. 7c. Similarly, OXPHOS had an attributed significant negative z-score indicating its down-regulation in ERR α KO and KI:KO hearts (Fig. 6b). However, with a $|FC| \geq 1.5$, this pathway would not be deemed deregulated as 83% of the contributing genes would be lost.

Extended Data Fig. 7c

10. For Supplementary Figure S3b, would recommend listing terms in order of descending number of DEGs with priority for KI:KO model to more easily observed which cardiac targets may be more affected.

Response:

We have updated Extended Data Fig 3b as suggested and shown below. Noteworthy, given our need to re-analyze the phosphoproteomics dataset (see response to point 7), all associated IPA analyses were performed with a more recent software version. As such, all other analyses were re-done with the same IPA version for consistency including the analysis involving the transcriptome data for Extended Data Figure S3b.

Extended Data Fig. 3b

b

IPA: Selected Cardiac-related Toxicological Functions						
Cardiac disease/function	ErbB2 KI vs WT		ERR α KO vs WT		KI:KO vs WT	
	p-value	# genes	p-value	# genes	p-value	# genes
Hypertrophy of heart	6.13E-04	29	3.21E-13	103	1.20E-11	121
Dilated cardiomyopathy	ns	13	3.65E-10	76	5.85E-10	92
Arrhythmia	1.84E-02	18	8.16E-11	74	2.71E-11	91
Myocardial infarction	5.01E-03	18	3.90E-09	64	1.39E-09	79
Left ventricular dysfunction	ns	12	3.55E-11	64	5.90E-10	74
Failure of heart	ns	9	1.33E-05	62	3.92E-04	71
Fibrosis of heart	ns	9	1.17E-04	49	1.80E-03	56
Coronary artery disease	ns	9	1.10E-03	41	1.27E-03	51
Enlargement of heart cells	2.38E-04	16	6.43E-07	42	7.50E-05	45
Congestive heart failure	ns	3	3.10E-05	32	1.32E-03	34
Damage of heart	ns	3	2.54E-06	22	2.26E-04	22

11. Pg 10, "...with 40% of top 15 distinguished metabolites annotated to lipid metabolism (eg acetyl-CoA, corticosterone) alongside metabolites to glycogen break down and glycolysis (pyruvate)." Should be state pyruvate is one of the top metabolites first (higher VIP score), though the majority of metabolites represent lipids.

Response:

Considering our need to re-analyze the phosphoproteomics data, in parallel we took the liberty to also update our metabolomics analysis by use of the current version of MetaboAnalyst (version 5 from version 3). By doing so, 50 additional metabolites were mapped by the software as compared to previously for a total of 352 of 383 DEMs identified across the groups vs WT that were mapped and used in the analyses by MetaboAnalyst (heatmap, PLS-DA, Random Forest classification). As such, the results differ slightly than from before. As suggested by the reviewer, we have modified the phrase regarding the PLS-DA results as follows, "Supervised clustering using partial least squares-discriminant analysis (PLS-DA) showed a similar group separation as to unsupervised clustering (Fig. 5b), highlighting ADP-ribose, pyruvate and AICAR as the most distinguishing metabolites (strongest VIP scores), though 40% of the top 15 were annotated to lipid metabolism (e.g., acetyl-CoA, corticosterone) (Extended Data Fig. 5a)."

Figure S5

12. Pg 13 “Intriguingly, a doxorubicin response...,” prior to this statement, there was no mention of the purpose for doxorubicin treatment. A brief summary of purpose should be included to assist transition.

Response:

We agree with the reviewer and modified the text as follows, “Given that mitochondria are a primary target of doxorubicin action, that loss of ERR α triggers mitochondrial dysfunction, and that both facets augment cardiotoxicity induced by ErbB2-targeted approaches, we surmised a strong inverse relationship between doxorubicin and ERR α signaling. Fittingly, a doxorubicin response was deemed activated exclusively in the ERR α KO-driven signature (Fig. 6c), suggesting that impaired ERR α function is a molecular facet of doxorubicin-driven cardiotoxicity. To explore this notion,...”.

13. Regarding minor edits:

- Please fix spelling of “remodelling” to “remodeling” and “signalling” to “signaling” throughout text of the manuscript.

Response:

These changes were done.

- *Breast cancer (BCa) has already been defined on pg 3 of the manuscript; please ensure it is not repeated again and continue the use of the acronym for manuscript continuity (ex. Pg 4). Similarly, DCM has been identified earlier in introduction, on pg 7 use acronym vs saying “dilated cardiomyopathy.”*

Response:

These changes were done.

- *Consider clarifying the background of KI:KO mice when discussing cross on pg 5 and also clarify the investigation of response in ERR α -KI and Erbb2 KO mice.*

Response:

We clarified the FVB background of the mice and that ErbB2 KI and ERR α KO mice were investigated in parallel with KI:KO mice in comparison to WT controls.

- *Please consistently use acronyms such as ERR α -KI vs ERR α -null to increase readability.**

Response:

We ensured to always use ERR α KO instead of the alternative ERR α -null. ErbB2 KI are always referred to as ErbB2 KI.

- *Supplementary Fig 2E- if possible would edit “GAP” to say “Gja1” to be consistent with the text (pg 6). If incorrectly identified, please clarify the appropriate site of reference.*

Response:

Actually, “GAP” in now Extended Data Fig. 2f refers to a biological term not a specific phosphoprotein. In the text, we indicate that Gja1 phosphosite alterations were attributed to several enriched biological terms including GAP by KEA2 analysis.

- *Figure legends should clearly indicate which statistical test was employed.*

Response:

The legends were modified accordingly.

REVIEWERS' COMMENTS:

Reviewer #1 (Remarks to the Author):

The authors have made a good faith effort to address my comments.

Though the initial design of the phospho-proteomic was a bit flawed (incidentally, there usually is sufficient material to do phospho-proteomics and expression proteomics on the same samples), the authors have revamped their statistical analysis, validated several findings by immunoblotting, and acknowledged the limitations of the study. The authors also were able obtain some (though not all) raw data from Metabolon to be deposited, which is commendable.

I have no further major concerns.

Minor point:

In the limitations part of the discussion, the authors mention that MS1 based quantitation is "semi-quantitative". That seems unnecessarily harsh and self-deprecating. MS1 quant may not be as accurate or sensitive as targeted quant, but it's still reasonably accurate and fairly robust when properly normalized. Now...spectral counting...that's semi-quantitative....

Reviewer #2 (Remarks to the Author):

The manuscript represents an excellent source of information about the impact of ERBB2 and estrogen receptor deficiencies, and its multi-omics approach provides a visualization from multiple points of view.

The authors addressed very well previous comments made by me and other reviewers.

A huge improvement of the manuscript was obtained by refining the data processing, and the addition of alternative techniques for result validation strengthen the results.

I consider that the figures and tables are satisfactory, the statistical methods are valid and correctly applied, also, the methods sufficiently documented to allow replication studies.

Reviewer #3 (Remarks to the Author):

The authors have addressed all concerns, comments and suggestions in a comprehensive fashion. The revised manuscript, figures and especially the discussion are remarkably improved and I recommend acceptance.

Response to reviewer

We thank the reviewers for their prompt and positive review of our revised manuscript. In response to comments by reviewer 1, we have eliminated one phrase of the limitations section as suggested.